# Digital restoration of the pectoral girdles of two Early Cretaceous birds and implications for early-flight evolution

Shiying Wang[1,2,3], Yubo Ma[4], Qian Wu[1,2,3], Min Wang[1,2]*, Dongyu Hu[5]*, Corwin Sullivan[4,6], Xing Xu[1,2,5,7]*

[1]Key Laboratory of Vertebrate Evolution and Human Origins, Institute of Vertebrate Paleontology and Paleoanthropology, Chinese Academy of Sciences, Beijing, China; [2]CAS Center for Excellence in Life and Paleoenvironment, Beijing, China; [3]University of Chinese Academy of Sciences, Beijing, China; [4]University of Alberta, Edmonton, Canada; [5]Shenyang Normal University, Shenyang, China; [6]Philip J. Currie Dinosaur Museum, Wembley, Canada; [7]Centre for Vertebrate Evolutionary Biology, Yunnan University, Kunming, China

*For correspondence:
wangmin@ivpp.ac.cn (MW);
hudongyu@synu.edu.cn (DH);
xingxu@vip.sina.com (XX)

**Competing interest:** The authors declare that no competing interests exist.

**Abstract** The morphology of the pectoral girdle, the skeletal structure connecting the wing to the body, is a key determinant of flight capability, but in some respects is poorly known among stem birds. Here, the pectoral girdles of the Early Cretaceous birds *Sapeornis* and *Piscivorenantiornis* are reconstructed for the first time based on computed tomography and three-dimensional visualization, revealing key morphological details that are important for our understanding of early-flight evolution. *Sapeornis* exhibits a double articulation system (widely present in non-enantiornithine pennaraptoran theropods including crown birds), which involves, alongside the main scapula-coracoid joint, a small subsidiary joint, though variation exists with respect to the shape and size of the main and subsidiary articular contacts in non-enantiornithine pennaraptorans. This double articulation system contrasts with *Piscivorenantiornis* in which a spatially restricted scapula-coracoid joint is formed by a single set of opposing articular surfaces, a feature also present in other members of Enantiornithines, a major clade of stem birds known only from the Cretaceous. The unique single articulation system may reflect correspondingly unique flight behavior in enantiornithine birds, but this hypothesis requires further investigation from a functional perspective. Our renderings indicate that both *Sapeornis* and *Piscivorenantiornis* had a partially closed triosseal canal (a passage for muscle tendon that plays a key role in raising the wing), and our study suggests that this type of triosseal canal occurred in all known non-euornithine birds except *Archaeopteryx*, representing a transitional stage in flight apparatus evolution before the appearance of a fully closed bony triosseal canal as in modern birds. Our study reveals additional lineage-specific variations in pectoral girdle anatomy, as well as significant modification of the pectoral girdle along the line to crown birds. These modifications produced diverse pectoral girdle morphologies among Mesozoic birds, which allowed a commensurate range of capability levels and styles to emerge during the early evolution of flight.

## Editor's evaluation

The authors provide new 3D fossil findings in *Sapeornis*, an avialan that lived during the Early Cretaceous period, a key node in our understanding of the evolution of avian flight. The functional reconstruction of two critical skeletal elements of the avian flight apparatus, the scapula and coracoid, enables the authors to hypothesize how the evolution of the scapula and coracoid enabled the modern avian wing stroke. The new 3D morphological reconstruction enables future integrative

studies of *Sapeornis* flight performance based on biomechanical, muscle physiological, and aerodynamic principles and is thus a key building block to inform our general understanding of the evolution of avian flight.

## Introduction

The evolution of powered flight in birds was one of the great transformations in vertebrate history and involved a suite of dramatic anatomical changes that were required to produce a functional flight apparatus (*Dudley and Yanoviak, 2011*; *Padian, 1985*; *Rayner, 1988*; *Videler, 2005*; *Burch, 2014*; *Jasinoski et al., 2006*). The pectoral girdle, a skeletal structure that connects the forelimb to the trunk, is a key component of the flight apparatus, and its function and evolutionary history have been extensively studied (*Baier et al., 2007*; *Bock, 2013*; *Novas et al., 2020*; *Senter, 2006*; *Burch, 2014*; *Jasinoski et al., 2006*; *Ostrom, 1976*). The morphology of the pectoral girdle in Late Cretaceous enantiornithine birds is well known from several three-dimensionally preserved specimens (*Atterholt et al., 2018*; *Chiappe and Walker, 2002*; *Chiappe et al., 2007*). However, most Early Cretaceous bird fossils are essentially two-dimensionally preserved as slab specimens, and accordingly do not offer a full anatomical picture of the flight apparatus, a limitation that greatly hinders studies of early flight. Here, we reconstruct the pectoral girdles of the non-ornithothoracine bird *Sapeornis chaoyangensis* (PMoL-AB00015) and the enantiornithine bird *Piscivorenantiornis inusitatus* (IVPP V 22582) using computed tomography and three-dimensional visualization. Our renderings are the first three-dimensional ones of the pectoral girdle for these two Early Cretaceous birds and reveal some important anatomical details for understanding pectoral girdle evolution. One main objective of our study was to better understand the evolution of the scapula-coracoid articulation and triosseal canal on the line to crown group birds because the form of the scapula-coracoid articulation has traditionally been used to distinguish between enantiornithines and euornithines (ornithuromorphs), whereas the nature of the triosseal canal has implications for the course of the tendon of M. supracoracoideus and thus for the mechanics of the upstroke during flight.

The pectoral girdle underwent dramatic changes in the shape, position, and orientation of each component element along the line to crown birds (*Xu, 2002*). In early-diverging theropods (*Figure 1A*), the scapula and coracoid lie obliquely on the ribcage with the anatomically cranial (anterior) edge of the coracoid significantly lower than the trunk vertebral column; the glenoid fossa is caudoventrally (posteroventrally) directed; the scapula has a large craniodorsally (anterodorsally) oriented acromion process; and the coracoid is a laterally facing semicircular plate with a small coracoid tubercle (called the biceps tubercle in some studies, and a precursor to the acrocoracoid process of birds). In early-diverging maniraptoriform theropods, the coracoid is somewhat biplanar, being divided by a line of deflection into two subtriangular areas that we term the sternal wing (called the distal ramus in *Xu, 2002*) and the scapular wing (called the proximal ramus in *Xu, 2002*) of the coracoid. In some maniraptoriform species, the line of deflection is marked by a ridge originating from the coracoid tubercle, on the coracoid's lateral surface. In other species, a distinct ridge is absent, and only the deflection itself defines the boundary between the scapular and sternal wings. The deflection causes the originally laterally facing lateral surface of the coracoidal sternal wing to be directed somewhat cranially (anteriorly) and ventrally. The scapular wing comprises the glenoid fossa, the scapular articulation, and the thin sheet normally housing the supracoracoid foramen, which accommodates the supracoracoid nerve. In early-diverging pennaraptorans (*Figure 1B, E, and F*), the glenoid of the scapulocoracoid is close to the trunk vertebral column and faces laterally; the scapula has a small acromion process; the scapular blade is nearly parallel to the trunk vertebral column, with its lateral surface tilted dorsally; and the coracoid has a large coracoid tubercle, a large sternal wing, and a small scapular wing. The originally lateral surface of the sternal wing faces cranially (anteriorly) and that of the scapular wing is directed craniodorsally, so that the coracoid more closely resembles an inverted 'L' than a semicircular plate in lateral view. In most avialans (*Figure 1C and G*), the glenoid fossa is dorsolaterally oriented; the scapular acromion process protrudes farther cranially; the scapular blade is twisted so that the anatomically lateral surface of the proximal portion faces dorsally, whereas that of the distal portion faces dorsolaterally; and the coracoid is a strut-like structure with several derived features (e.g., the coracoid tubercle is enlarged to form the acrocoracoid process; the sternal wing is elongated in a craniodorsal-caudoventral direction; the scapular wing is highly reduced, bringing the

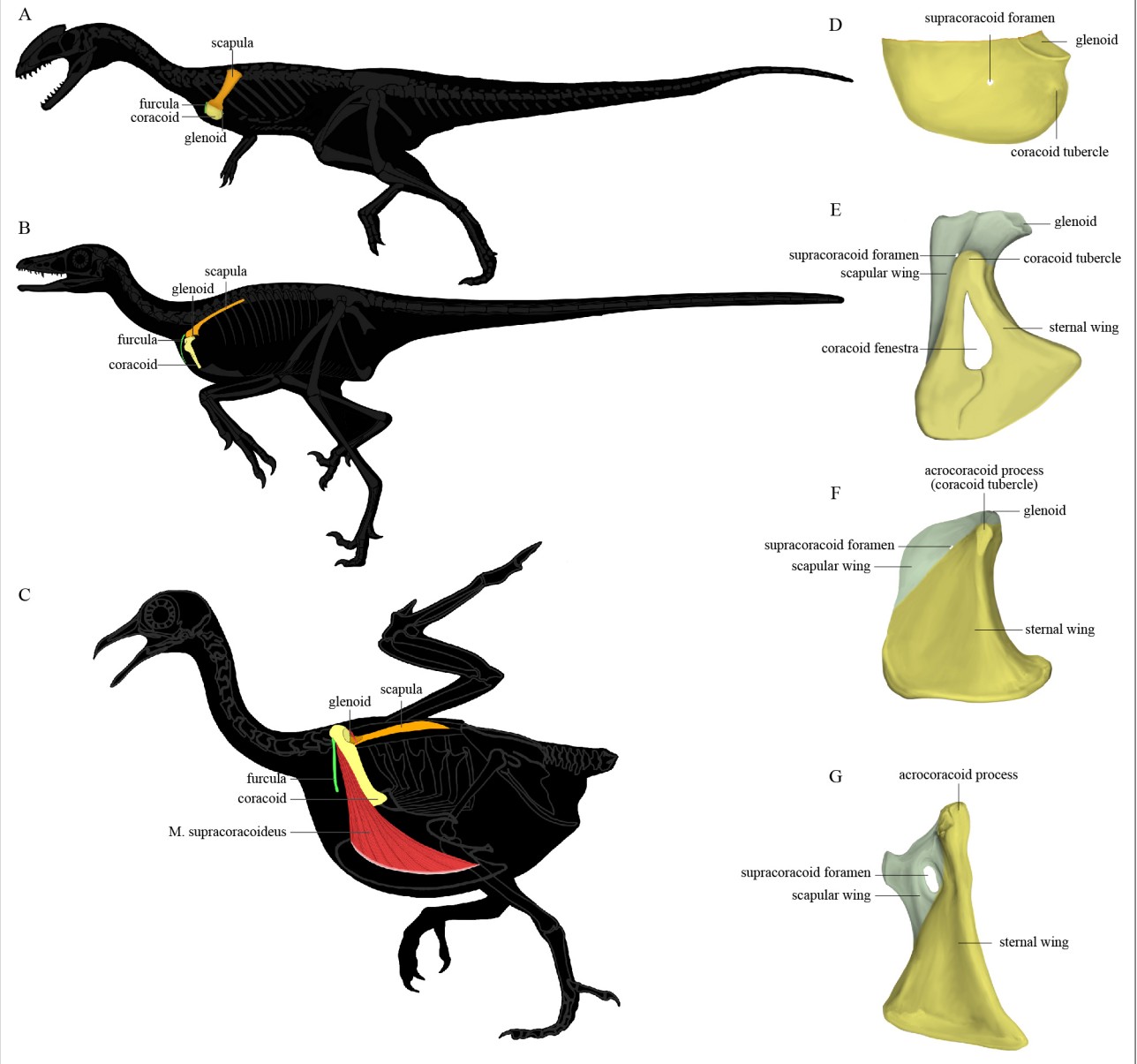

**Figure 1.** The position of the pectoral girdle and the form of the coracoid in different theropod groups. (**A–C**) Skeletal silhouettes showing the anatomical position of the pectoral girdle in (**A**) the early-diverging theropod *Coelophysis*, (**B**) the early-diverging pennaraptoran *Microraptor,* and (**C**) the modern bird *Columba*. The M. supracoracoideus is illustrated in (**C**) but typically covered by the M. pectoralis, which is not illustrated. (**D–G**) Illustrations of the left coracoids of (**D**) *Coelophysis* (modified from *Tykoski, 1998*), (**E**) the early-diverging pennaraptoran *Sinornithosaurus* (modified from *Xu et al., 1999*), (**F**) the early-diverging avialan *Archaeopteryx* (modified from *Wellhofer et al., 2009*), and (**G**) the early-diverging avialan *Jeholornis* (based on STM 2-49 and IVPP V 13886). Coracoid of *Coelophysis* in lateral view, coracoids of other taxa in ventral view.

scapular articulation and glenoid fossa extremely close to the sternal wing; and in many species, the scapular wing gives rise to a procoracoid process). In species with an extremely small scapular wing, the supracoracoid foramen is either absent or located in the sternal wing. *Figure 1* illustrates the key morphological features of the pectoral girdle in different theropod groups and the terms used in this article, though we admit that a complete evolutionary picture of the theropod pectoral girdle has yet to be presented and there are different opinions on what terms should be used for certain structures (*Xu, 2002*; *Novas et al., 2021*).

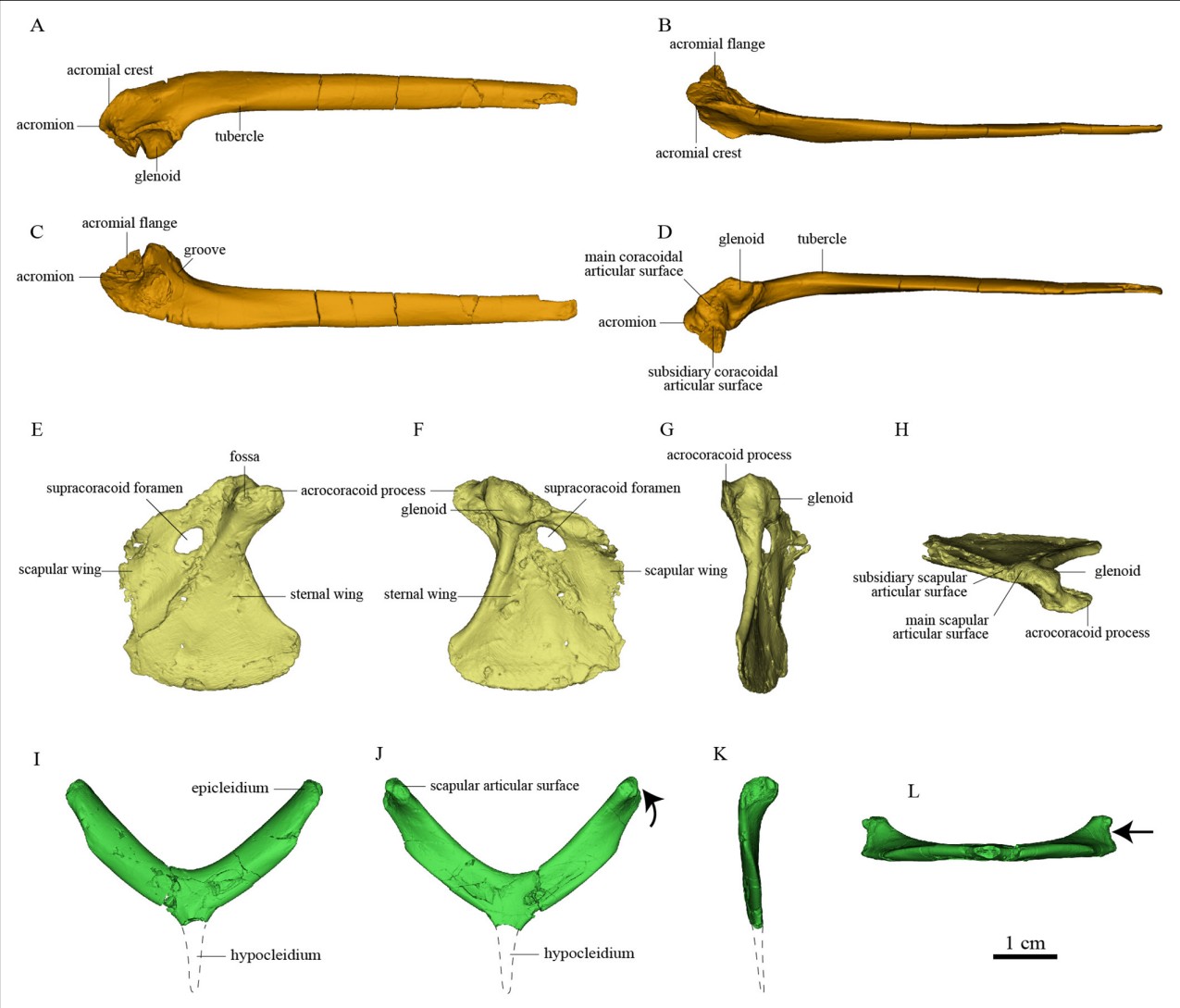

**Figure 2.** Pectoral girdle bones of *Sapeornis chaoyangensis* PMoL-AB00015. (**A–D**) Left scapula in lateral, dorsal, medial (costal), and ventral views. (**E–H**) Left coracoid in ventral, dorsal, lateral, and cranial views. (**I–L**) Furcula in cranial, caudal, lateral, and ventral views. The black arrows in (**J**) and (**L**) indicate the concave surface for the tendon of M. supracoracoideus.

## Results

The nearly complete pectoral girdle elements of *S. chaoyangensis* PMoL-AB00015 and *P. inusitatus* IVPP V 22582 have been three-dimensionally rendered in detail based on computed tomography scan data (*Figure 2*, *Figure 3*, and Figure5). However, the bones have been compressed during fossilization, so the renderings do not precisely capture the original morphology. Originally, the furcula of *Sapeornis* PMoL-AB00015 was probably slightly curved caudally (despite being straight in our rendering), and the angle between the scapular and sternal wings of the coracoid that contacted the scapula and the sternum was probably smaller than in our rendering. Because of these distortions, the pectoral girdle of *Sapeornis* based on digitally articulating our uncorrected renderings is characterized by a larger distance between the two coracoids, and a more ventrally oriented glenoid fossa, than would have been present in the skeleton of the living animal (*Gao et al., 2012*). However, these errors do not affect our major conclusions.

### Osteology of the pectoral girdle of *Sapeornis* PMoL-AB00015

The cranial part of the left scapula is curved medially and ventrally. The scapular blade is slightly twisted about its longitudinal axis, so that the anatomically lateral surface of the proximal portion

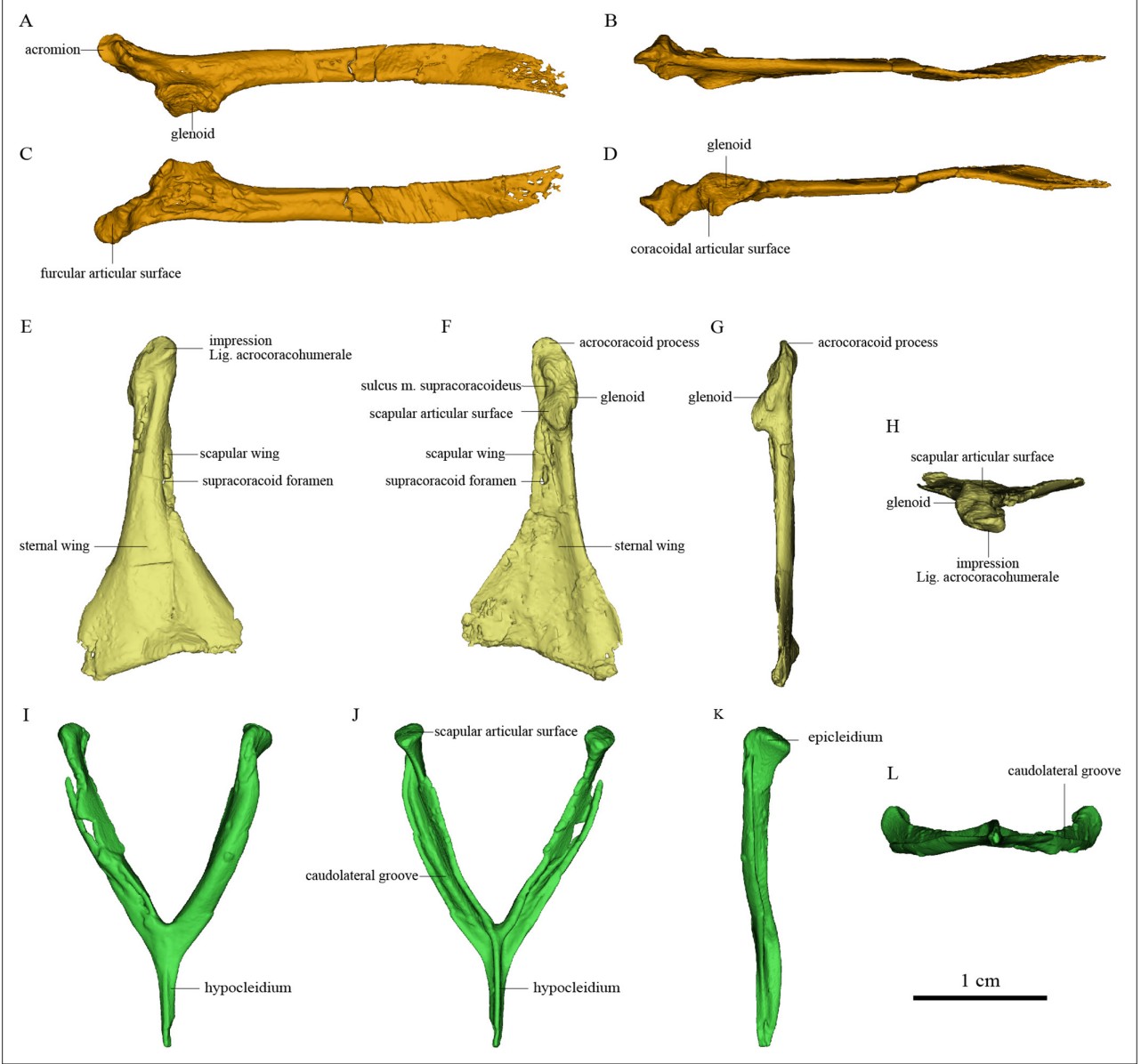

**Figure 3.** Pectoral girdle bones of *Piscivorenantiornis inusitatus* IVPP V 22582. (**A–D**) Left scapula in lateral, dorsal, medial (costal), and ventral views. (**E–H**) Right coracoid in ventral, dorsal, lateral, and cranial views. (**I–L**) Furcula in cranial, caudal, left, and ventral views.

faces dorsally, whereas that of the distal portion faces dorsolaterally as in extant birds (*Figure 2*). The acromion is short. As in the dromaeosaurid *Rahonavis* (*Forster et al., 2020*), a broad flange protrudes medially from the acromion (*Figure 2*), adding to a previously known set of derived similarities shared by *Rahonavis* and some early-diverging avialans (*Novas et al., 2018*). As in *Jeholornis* (*Zhou and Zhang, 2003a*), the scapular glenoid fossa faces mainly ventrally but also slightly laterally, showing more lateral deflection than in deinonychosaurs (e.g., *Sinovenator* and *Rahonavis*) (*Forster et al., 2020*) and *Archaeopteryx* (*Zhou and Zhang, 2003a*). The articular surface for the coracoid consists of two parts: a deeply concave main surface situated craniomedial to the glenoid facet on the ventral surface of the scapula and a slightly concave subsidiary surface positioned more craniomedially (*Figure 4*). A weak tubercle lies on the ventrolateral margin of the scapular blade, possibly representing the muscle insertion site for M. subscapulare (*Figure 2*; *Gianechini et al., 2018*). A short and shallow groove (*Figure 2*) on the medial surface of the scapula, close to the glenoid fossa and subparallel to the ventral margin, probably represents another muscle insertion site.

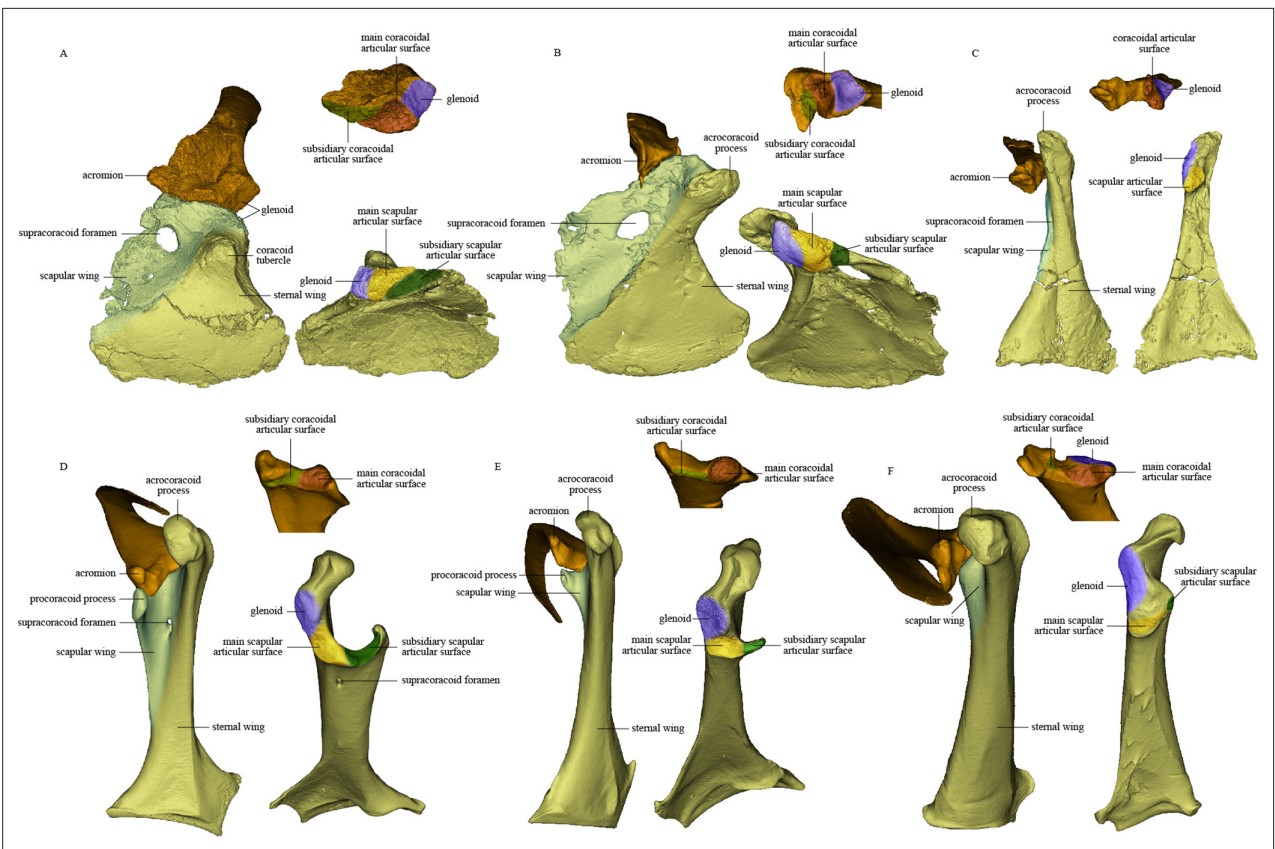

**Figure 4.** Comparison of scapula and coracoid morphology across various paravian taxa. Each panel shows articulated left scapula and coracoid in ventral view (on left) and opposing articular surfaces of left scapula and coracoid (on right, with cranial direction toward top of figure for both scapula and coracoid). (**A**) *Sinovenator changii* (mirrored), (**B**) *Sapeornis chaoyangensis*, (**C**) *Piscivorenantiornis inusitatus*, (**D**) *Tyto alba*, (**E**) *Egretta garzetta*, and (**F**) *Pavo muticus*.

The coracoid is in general similar to that of non-avialan pennaraptorans in having a large scapular wing, with the glenoid fossa and scapular articulation well separated from the sternal wing. The caudal margin of the sternal wing is slightly convex and lacks a visible articular facet for the sternum (*Figure 2*), rather than being straight to concave and bearing a sternal facet as in *Jeholornis* (*Wang et al., 2020a*) and most ornithothoracines (*Atterholt et al., 2018*; *Wang and Zhou, 2017a*). The lack of a sternal facet lends further support to previous inferences that an ossified sternum is genuinely absent in this early pygostylian lineage (*Zheng et al., 2014*). In living birds, the ossified sternum provides the major attachments for M. supracoracoideus and M. pectoralis; while in *Sapeornis*, the large coracoid may have served to provide a proximal attachment surface for these muscles (*Zheng et al., 2014*). On the ventral surface of the coracoid, a ridge extends from the acrocoracoid process to the distomedial corner of the bone, clearly demarcating the scapular and sternal wings of the coracoid (*Figure 2*) as in some maniraptoriform theropods (e.g., *Sinornithosaurus*) (*Xu, 2002*). The sternal wing is short, having a ratio of cranial-caudal length to medial-lateral width at the caudal margin of only 1.17. This is close to the value for *Archaeopteryx* (~1.15), but differs from those for the more elongated sternal wings of *Jeholornis*, *Confuciusornis*, and most ornithothoracines (generally >1.5). The scapular wing is large, as in most non-avialan pennaraptorans, but in contrast to the reduced scapular wing seen in such dromaeosaurids as *Microraptor* and in the avialans *Archaeopteryx* and *Jeholornis* (*Wang et al., 2020aWang et al., 2020a*; *Wellnhofer et al., 2009*; *Xu et al., 2003*). In most euornithines, the scapular wing is likewise small, and curves ventrally and gives rise to a narrow procoracoid process at the craniomedial corner (e.g., *Egretta garzetta*; *Figure 4E*). In some euornithine species (e.g., *Tyto alba*; *Figure 4D*), however, the scapular wing is thin and relatively large, somewhat similar to the condition in non-enantiornithine pennaraptorans. As in *Jeholornis* (*Wang et al., 2020a*), the supracoracoid foramen (*Figure 2*) is large and positioned within the scapular wing.

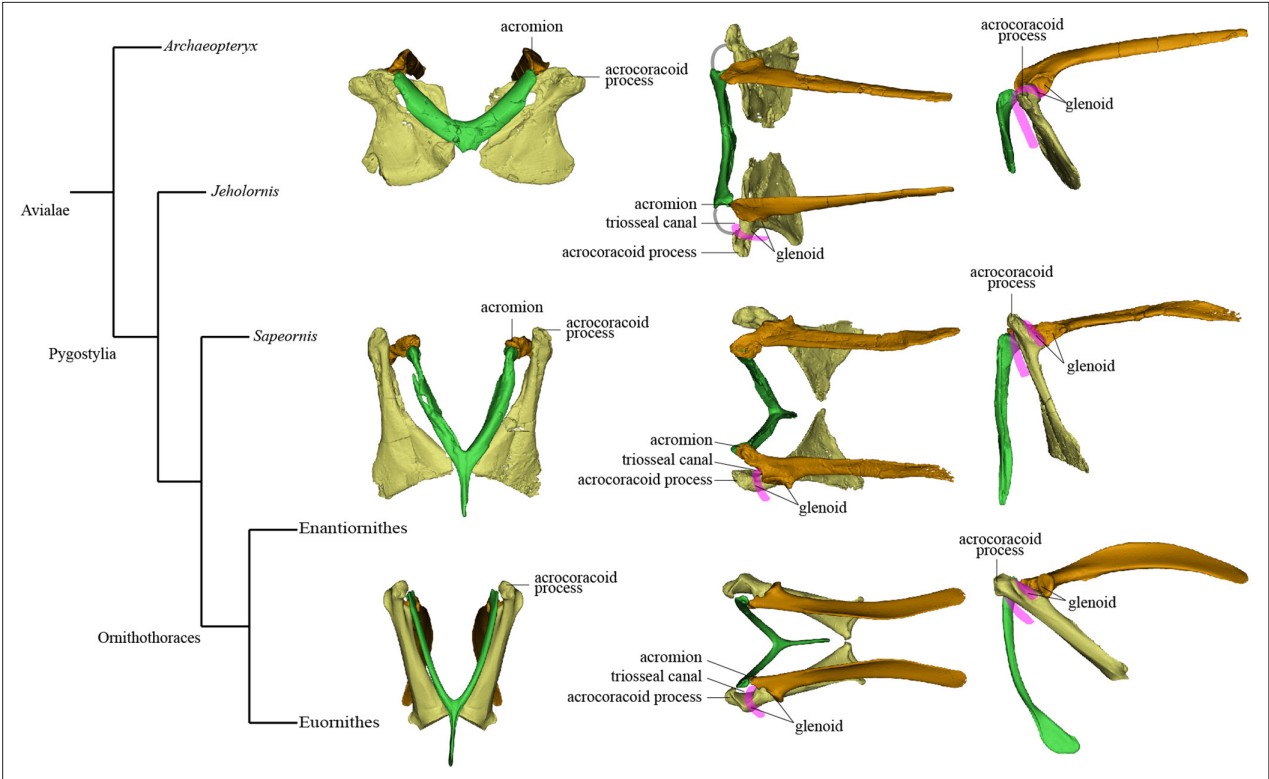

**Figure 5.** Simplified phylogeny with hypothetical steps in pectoral girdle evolution. The pectoral girdles of *Sapeornis chaoyangensis*, *Piscivorenantiornis inusitatus,* and *Pavo muticus* (from top to bottom) are shown in cranial, dorsal, and left lateral views. The pink lines in the dorsal and lateral views represent the tendon of M. supracoracoideus, and the gray line in the dorsal view of the *Sapeornis* rendering represents the coracoclavicular ligament that connects the coracoid and furcula. Phylogenetic framework following *Wang et al., 2018a*.

The acrocoracoid process is short and blunt, and extends slightly above the midpoint of the coracoidal glenoid fossa as in *Jeholornis* and *Confuciusornis* (*Turner et al., 2012Wang and Zhou, 2018bWang and Zhou, 2018b*; *Wang et al., 2020a*; *Zhou and Zhang, 2003b*; *Zhou and Zhang, 2003b*). In enantiornithines (*Panteleev, 2018*; *Chiappe and Walker, 2002*), the acrocoracoid process extends slightly above the dorsal margin of the coracoidal glenoid fossa. In most euornithines (e.g., *Figure 4D–F*), this process is proportionally longer and extends much further beyond the glenoid than in non-euornithine birds. In non-avialan theropods and *Archaeopteryx*, by contrast, the acrocoracoid process (frequently described as coracoid tubercle or biceps tubercle) is located cranioventral to the glenoid fossa (*Mayr et al., 2005*; *Novas et al., 2021*). The acrocoracoid process of *Sapeornis* forms a shelf-like structure projecting dorsally, cranially, and laterally from the lateralmost part of the coracoid, a condition not known in other birds. A small shallow fossa with an irregular surface, located at the medial end of the cranioventral surface of the acrocoracoid process (*Figure 2*), may have provided an attachment point for a coracoclavicular ligament connecting the coracoid and furcula (*Figure 5*). In some volant extant birds, by contrast, the coracoclavicular ligament attaches to the cranial edge of the medial surface of the acrocoracoid process (*Ghetie, 1976*).

The glenoid fossa is on the craniolateral corner of the dorsal face of the coracoid and wraps onto the cranial margin. The scapular articular surface is situated entirely on the coracoid's cranial margin and is not separated from the coracoidal glenoid fossa by any distinct border. The coracoidal glenoid fossa is weakly concave and faces dorsocaudally and only slightly laterally, whereas the lateral tilt of the fossa is greater in late-diverging birds (*Figure 4*).

Like the opposing surface on the scapula, the scapular articular surface on the coracoid is divided into main and subsidiary parts. The former is weakly convex, to match the concave main articular surface for the coracoid on the scapula, whereas the latter extends along the cranial margin of the scapular wing of the coracoid and contacts the subsidiary coracoidal articular surface on the scapula.

The robust, craniocaudally compressed furcula presumably would not have been as flexible as those of volant extant birds, which have mediolaterally compressed rami and in at least some cases act as a spring during flapping flight (*Boggs et al., 1997*; *Nesbitt et al., 2009*). The omal third of the ramus is slightly narrower than the remainder and terminates in a blunt epicleidium. In dorsal view, the epicleidium is twisted laterally by about 80° relative to the ramus. Similar torsion can also be observed in some other early birds (e.g., *Confuciusornis*) and in some deinonychosaurs (e.g., *Halszkaraptor*, *Buitreraptor*) (*Cau et al., 2021*; *Gianechini et al., 2018*). The lateral surface of the epicleidium is concave, making the epicleidium C-shaped in dorsal view. This concave surface possibly accommodated the tendon of the supracoracoideus muscle, which would have passed through the partially closed triosseal canal and over the small acrocoracoid process to reach its insertion on the humerus (*Figure 5*). As in *Confuciusornis* (*Wu et al., 2020*) and *Xiaotingia* (*Xu et al., 2011*), a small caudal projection is located on the medial side of the epicleidium and probably articulated with the acromion of the scapula. The short, slender hypocleidium is broken away, but the probable outline of the hypocleidium is indicated in *Figure 2I–K* based on another specimen of *Sapeornis* (IVPP V 19058).

## Osteology of the pectoral girdle of the *P. inusitatus* IVPP V 22582

The pectoral girdle is closely comparable in morphology to those of other enantiornithines (*Hu et al., 2015a*; *Zhang et al., 2014*; *Chiappe and Walker, 2002*). The long and robust acromion of the scapula is separated by a neck from the coracoidal articular surface. The cranial part of the medial surface of the acromion forms a flat articular surface for the furcula, as in many other enantiornithines (e.g., *Rapaxavis* and *Halimornis*) (*Chiappe et al., 2002*; *O'Connor et al., 2011*). The subtriangular scapular glenoid fossa faces more laterally than in *Sapeornis* and non-avialan theropods (*Brusatte et al., 2013*; *Funston et al., 2020*), although the orientation of the fossa is nevertheless also somewhat ventral. The slightly concave coracoidal articular surface lies cranial and medial to the scapular glenoid fossa, and is nearly perpendicular to the latter. This surface corresponds to the main coracoidal articular surface on the scapula of non-enantiornithine pennaraptorans, and that the subsidiary coracoidal articular surface is absent. The scapular blade is curved ventrally and tapered caudally.

The coracoid is subtriangular in dorsal or ventral view. The acrocoracoid process is rounded and minimally developed, as is typical in enantiornithines (*Atterholt et al., 2018*; *Zhang et al., 2014*; *Panteleev, 2018*). The coracoidal glenoid fossa is craniocaudally elongated and faces caudally and somewhat dorsolaterally. The slightly convex scapular articular surface is smaller than the coracoidal glenoid fossa, and is situated dorsomedial to the latter as in other ornithothoracines. In *Sapeornis* and non-avialan theropods, the scapular articular surface is proportionally larger and situated directly medial to the coracoidal glenoid fossa. This surface corresponds to the main scapular articular surface on the coracoid of non-enantiornithine pennaraptorans, and that the subsidiary scapular articular surface is absent. On the medial side of the glenoid fossa, between the acrocoracoid process and scapular articular surface, lies a small incisure that is present in most enantiornithines (*Hu et al., 2015bHu et al., 2015b*; *Panteleev, 2018*; *Wang et al., 2016d*). This structure is identified as the sulcus M. supracoracoideus, through which the M. supracoracoideus tendon passed (*Hu et al., 2012*; *Panteleev, 2018*). A large impression (*Figure 3*) associated with attachment of Lig. acrocoracohumerale is located on the cranioventral surface of the acrocoracoid process, above the level of the ventral margin of the glenoid fossa, and faces cranially and slightly laterally. In extant birds, by contrast, the equivalent feature is located well craniodorsal to the glenoid fossa and faces more laterally. The sheet-like component of the scapular wing of the coracoid, termed the 'medial crest' by *Panteleev, 2018*, is extremely narrow and terminates at the base of the scapular articular surface. The loss of the subsidiary scapular articular surface is connected to the reduction of the coracoid's scapular wing. The scapular wing is perforated by a small supracoracoid foramen, as in most enantiornithines (*Atterholt et al., 2018*; *Chiappe et al., 2007*; *Panteleev, 2018*; *Wang et al., 2016a*). The neck of the sternal wing is proportionally shorter than the equivalent structure in most extant birds (*Panteleev, 2018*).

The furcula is robust and generally Y-shaped, with an interclavicular angle of only about 50° and a long hypocleidium, as in most enantiornithines (*Hu et al., 2015a*; *Zhang et al., 2014*). The epicleidium is expanded both craniocaudally and mediolaterally to form a dorsally facing articular facet for the coracoid, as in other enantiornithines (*Atterholt et al., 2018*). The midshaft of each ramus is 'L' shaped in cross section owing to the presence of a deep caudolateral groove, another characteristic of Enantiornithes (*Atterholt et al., 2018*; *Chiappe et al., 2007*; *Close et al., 2010*). The omal part of this

groove faces somewhat laterally due to torsion of the ramus and tapering of the cranial surface of the ramus from the lateral side. The groove extends ventrally to the end of the hypocleidium, producing a high keel on the caudal surface of the hypocleidium between the right and left grooves. The bilaterally compressed form of the hypocleidium is shared with many enantiornithines (*Wang et al., 2014*; *Wang et al., 2020b*), but differs from the craniocaudal compression seen in some oviraptorosaurs, some troodontids, and *Sapeornis* (*Nesbitt et al., 2009*; *Xu and Norell, 2004*).

## Discussion

*S. chaoyangensis* PMoL-AB00015 and *P. inusitatus* IVPP V 22582 provide significant new information on the pectoral girdle morphology of these two Early Cretaceous birds. This information bears, in particular, on the following issues pertaining to the early evolution of the avialan pectoral girdle.

### Morphology of the scapula-coracoid articulation, the scapular wing of the coracoid, and the procoracoid process

The scapula-coracoid articulation is variable in morphology among pennaraptoran theropods. In non-avialan pennaraptorans, the structure of the articulation between the scapula and coracoid is not well known. This is mainly because the two elements tend to fuse, albeit normally at a relatively late ontogenetic stage, to form a scapulocoracoid. For example, the scapula and coracoid are tightly sutured together, but not fused, in a juvenile specimen of *Velociraptor mongoliensis* (MPC-D100/54), and a fused scapulocoracoid is seen in an adult specimen (IGM 100/986) (*Hone et al., 2012*; *Norell and Makovicky, 1999*). Complete fusion of the scapulocoracoid, leaving no visible suture, is the usual condition in adult individuals (e.g., *Citipati osmolskae* IGM 100/1004, *Anzu wyliei* CM 78 001) (*Lamanna et al., 2014*; *Norell et al., 2018*). Among non-ornithothoracine avialans, all known jinguo-fortisids, and all known confuciusornithiforms except a single subadult *Eoconfuciusornis zhengi* specimen (IVPP V 11977), possess a fused scapulocoracoid (*Wang et al., 2018a*; *Wu et al., 2021*). The occurrence of scapula-coracoid fusion in *Archaeopteryx* is controversial, but the scapula and coracoid are separately preserved in some specimens (*Kundrát et al., 2018*; *Mayr et al., 2005*; *Wellnhofer et al., 2009*; *Wu et al., 2021*). In sapeornithiforms and jeholornithiforms, the scapula and coracoid are not fused (*Zhou and Zhang, 2003b*; *Zhou and Zhang, 2003b*). Among ornithothoracines, a fused scapulocoracoid is known only in flightless paleognaths (*Wu et al., 2021*).

In several deinonychosaurs (e.g., *Sinovenator* and *Rahonavis*), the scapula and coracoid not only fail to fuse but remain loosely rather than tightly connected, contacting one another via smooth articular surfaces rather than via a firm interdigitating suture as in other non-avialan pennaraptorans (*Forster et al., 2020*). In *Sinovenator* (*Figure 4A*), the glenoid fossa of the coracoid is smaller than the scapular articular surface as in other non-avialan theropods. The reverse is true in crown group birds, and also in many stem birds in which this anatomical region is well known, such as *Piscivorenantiornis* (*Figure 3*), *Elsornis* (*Chiappe et al., 2007*), *Mirarce* (*Atterholt et al., 2018*), and *Gansus* (*Wang et al., 2016c*). The scapular articular surface is located on the scapular wing of the coracoid and consists of two parts: a broadly convex main articular surface and a flat to shallowly concave subsidiary articular surface. The corresponding coracoidal articular surfaces on the scapula are both concave. The main and subsidiary articulations together form what is termed here a double articulation between the scapula and the coracoid. *Bambiraptor* also has a double articulation, in which the smaller, shallowly convex subsidiary articular surface on the coracoid is located mediodorsally to the flat main articular surface and is separated from the latter by a low ridge. The corresponding coracoidal articular surfaces on the scapula are shallowly concave, as in *Sinovenator*. This implies a gap between the main articular surfaces on the scapula and coracoid, which in life was presumably filled with cartilage. The dromaeosaurid *Rahonavis* possesses a double articulation, with a slightly concave main articular surface for the coracoid as in *Sinovenator* (*Forster et al., 2020*). As described above, the scapula-coracoid articulation of *Sapeornis* resembles that of *Sinovenator* in that the coracoid has a convex main scapular articular surface (matching the scapula's concave main coracoidal articular surface) and a subsidiary articular surface that would have contacted the cranioventral margin of the acromion process of the scapula. *Sapeornis* thus has a double articulation, consistent with the plesiomorphic overall shape of the scapulocoracoid, as in non-avialan pennaraptorans. Avialans other than enantiornithines have a broadly uniform type of scapula-coracoid articulation, although some morphological variation is

present. In *Jeholornis* and *Fukuipteryx,* the coracoid has a concave main articular surface for the scapula (*Imai et al., 2019*; *Turner et al., 2012*), a feature that has been proposed as an apomorphy of the Euornithes (Ornithuromorpha) in previous studies (*Wang and Zhou, 2017a*). In most euornithines, the coracoid indeed has a deep cotyla that receives a corresponding convexity on the scapula (*Figure 4E*). In some crown birds, however, the main scapular articular surface on the coracoid is flat to slightly convex (*Figure 4D and F*), as in many enantiornithine birds and in some non-avialan theropods, such as *Sinovenator*. In most euornithines and *Jeholornis*, the subsidiary articular surface for the scapula is situated partly on the procoracoid process, which is a projection of the dorsomedial margin of the small scapular wing of the coracoid.

Enantiornithines have a strikingly different scapula-coracoid articulation from other pennaraptorans. The presence of a single, convex articular surface, fitting into a cotyla on the cranial end of the scapula, has been widely accepted as a unique feature of the coracoid of enantiornithine birds (*Chiappe and Walker, 2002*; *Panteleev, 2018*; *Wang and Zhou, 2019*). Conversely, modern birds have been considered to display the opposite condition, with the main scapular articular surface on the coracoid being concave and that on the scapula convex. This purported discrepancy is the source of the clade name Enantiornithes, meaning 'opposite birds.' As mentioned above, however, a convex scapular articular surface is also seen on the coracoids of some non-avialan theropods, such as *Sinovenator*, and in some crown birds that are secondarily evolved (*Mayr, 2021*), though the convexity is less prominent in these taxa than in late-diverging enantiornithine birds. Furthermore, the scapular articular surface on the coracoid is shallowly concave in some enantiornithines, such as pengornithids (e.g., IVPP V 18687 and V 18632). Consequently, the 'opposite'-type scapula-coracoid articulation is neither present in all enantiornithine birds nor unique to the Enantiornithes. However, our study indicates that the enantiornithine scapula-coracoid articulation is indeed unique, but for a different reason: extreme reduction of the scapular wing of the coracoid and consequent loss of the subsidiary articular surface for the scapula in all enantiornithines (including pengornithids). This results in a single articulation in enantiornithines, in which the coracoid bears only one spatially restricted articular surface for the coracoid. The single articulation of enantiornithines is thus smaller in area, as well as morphologically simpler, than the double articulation of other pennaraptorans.

Combining anatomical details revealed by this study with information from the literature, three important modifications to the scapula-coracoid articulation may be inferred to have occurred among avialans: (1) loss of fusion between the scapula and coracoid in the majority of adult avialans, (2) displacement of the main articular surface for the scapula to a position extremely close to the base of the acrocoracoid process in a clade comprising *Jeholornis* and pygostylians (reversed in *Sapeornis* to the primitive condition of having the main articular surface relatively distant from the acrocoracoid process), and (3) establishment of the unique single articulation by extreme reduction of the scapular wing of the coracoid in enantiornithine birds. All non-enantiornithine pennaraptorans, including crown birds, have a double articulation connecting the scapula and coracoid, and in the majority of non-enantiornithine birds, a procoracoid process is present to buttress the double articulation and contribute to the triosseal canal.

## Architecture of the triosseal canal

The triosseal canal facilitates powered flapping flight in modern birds by forming a passage to admit the tendon of M. supracoracoideus, a muscle that contributes to humeral elevation and longitudinal rotation (*Baumel and Witmer, 1993*; *Poore et al., 1997*). However, the name 'triosseal canal' is misleading given the variable composition of this structure in living birds (*Livezey and Zusi, 2006*). The annotation provided for the triosseal canal by *Baumel and Witmer, 1993* explicitly described variation across taxa in the canal's architecture. In most extant birds, the furcula, coracoid, and scapula all participate (hence the name of the canal) in forming a fully enclosed bony passage. Typically, the epicleidium of the furcula forms the craniomedial wall of the triosseal canal, the acrocoracoid process of the coracoid forms the lateral wall, and the procoracoid process of the coracoid and the cranial margin of the acromion of the scapula form the caudomedial wall. However, in many extant birds (e.g., *Phalacrocorax capillatus* and *E. garzetta*), the bony triosseal canal is only partially closed, in that the furcula lacks a bony contact with the scapula but is bound to the latter by Lig. scapuloclaviculare dorsale (*Baumel and Witmer, 1993*). In *Rhea*, the Lig. acrocoracoacromiale bridges the tendon of M. supracoracoideus, contributing to a what is functionally a 'triosseal canal' (*Novas et al., 2021*). In

certain other birds, a closed bony canal is formed by the coracoid and scapula only, with no contribution from the furcula, or even formed by the coracoid alone via an ossified bridge connecting the acrocoracoid and procoracoid processes (e.g., *Upupa epops* and *Columba livia*) (*Baumel and Witmer, 1993*). Therefore, the triosseal canal is not necessarily formed by all three pectoral elements and is not necessarily a fully enclosed bony passage. Also, the procoracoid process is absent in certain volant crown birds that possess a triosseal canal, including *Pavo muticus* (*Figure 4*) and *Colius striatus* (*Mayr, 2021*), as well as in the Late Cretaceous galliform-like genus *Palintropus* (*Longrich, 2009*). Accordingly, the procoracoid process cannot be considered an essential constituent of the triosseal canal.

Among stem birds, a triosseal canal is widely accepted as present in early-diverging euornithines (*Mayr, 2017*; *Wang et al., 2016b*; *Zhou and Wang, 2017*). In most euornithine specimens (e.g., *Yixianornis* and *Gansus*) (*Clarke et al., 2006*; *Wang et al., 2016c*), the acrocoracoid process is medially hooked and a prominent procoracoid process is present, features that suggest the existence of a typical, fully enclosed bony triosseal canal formed by the scapula, coracoid, and furcula. Many previous studies have denied the presence of a triosseal canal in non-euornithine birds because of the lack of a long medially hooked acrocoracoid process and a procoracoid process (*Novas et al., 2021*; *Wang and Zhou, 2017a*). Although a small and pointed procoracoid process has been reported in *Protopteryx* (*Zhang and Zhou, 2000*; *Chiappe et al., 2020*), this cannot be confirmed in the provided figures and preservation of the omal region is poor. Other studies have argued that a triosseal canal is present in enantiornithines, albeit based on limited evidence (*Kurochkin et al., 2013*; *Zhang and Zhou, 2000*). *Mayr, 2017* argued that the tendon of M. supracoracoideus ran along the medial side of the acromion in enantiornithine birds, rather than along the lateral side as in crown birds, which would imply that the supracoracoideus pulley system was differently configured in enantiornithines than in euornithines.

Our renderings of the pectoral girdles of *Sapeornis* and *Piscivorenantiornis* indicate the presence of a partially enclosed triosseal canal in these early stem birds. In *Sapeornis,* the lateral wall of the triosseal canal is formed by the acrocoracoid process, the medial wall by the furcula, and the caudomedial wall by the scapular wing of the coracoid and the cranial margin of the acromion of the scapula. In these respects, the triosseal canal of *Sapeornis* has essentially the same structure as in most extant flying birds. In *Piscivorenantiornis*, the scapular wing of the coracoid is extremely reduced and lacks a procoracoid process. The caudomedial wall of the triosseal canal is formed by the cranioventral margin of the long acromion process of the scapula and the floor of the sulcus M. supracoracoideus of the coracoid, as in some extant birds that lack a procoracoid process, for example, *Corvus corax* and *P. muticus* (*Figure 4*). Both *Sapeornis* and enantiornithines have features of the pectoral girdle (e.g., dorsolaterally projecting acrocoracoid process in *Sapeornis*, and elongate acromion process and widely spaced acrocoracoid processes in enantiornithines) that imply the lack of a coracoid-furcula contact (*Mayr, 2017*; *Novas et al., 2021*). Absence of such a contact is a plesiomorphic feature, widely observed among non-avialan theropods (*Currie and Zhiming, 2001*; *Klingler, 2020*; *Lü, 2003*). Thus, *Sapeornis* and enantiornithines have only a partially enclosed bony triosseal canal, though a ligament could have completed the enclosure of the canal as in some modern birds (*Ghetie, 1976*). Absence of the cranial bony wall of the triosseal canal presumably would not affect the upstroke in these birds as the acrocoracoid process redirects the tendon of M. supracoracoideus (*Baumel and Witmer, 1993*).

## Pectoral girdle evolution and early flight

Mapping major aspects of pectoral girdle morphology onto an avialan phylogenetic tree suggests that three important evolutionary steps can be defined along the line to the modern flight apparatus and reveals some distinctive features characterizing certain avialan clades (*Figure 5*). Step I occurs at the base of the clade comprising *Jeholornis* and pygostylians, and involves torsion and elongation of scapular blade (ratio of scapular length to femoral length about 0.9, compared to 0.68–0.81 in *Archaeopteryx* [*Rauhut et al., 2018*], 0.68 in *Anchiornis* [*Hu et al., 2009*], and 0.58–0.73 in several dromaeosaurids [*Burnham et al., 2000*; *Hwang et al., 2002*; *Makovicky et al., 2005*]); elongation of sternal wing of coracoid (ratio of cranial-caudal length of sternal wing to medial-lateral width of caudal margin of sternal wing greater than 1.5; reversed to primitive condition in *Sapeornis*) (*Zhou and Zhang, 2003b*) shifting of scapular articular surface and glenoid fossa of coracoid to position extremely close to base of acrocoracoid process; reduction in area occupied by scapula-coracoid articulation compared to condition in non-avialan theropods; elongation of acrocoracoid process,

which is situated at dorsoventral level of coracoidal glenoid fossa; reduction in distance between left and right coracoids, indicating a relatively narrow and deep chest; reduction in angle between scapula and coracoid (*Novas et al., 2021*) and establishment of partially closed triosseal canal. Step II occurs at the base of Ornithothoraces and involves cranial extension and thickening of the acromion process (*Novas et al., 2021*) reduction of the interclavicular angle (generally to less than 65°) (*Hu et al., 2015b*) and reduction of the supracoracoid foramen. Step III occurs in early-diverging Euornithes and involves increased downward curving of caudal end of scapular blade (*O'Connor et al., 2016*) shifting of glenoid fossa of scapula onto the external surface of bone, causing the fossa to face dorsolaterally (*Wellnhofer et al., 2009*) appearance of the procoracoid process on the scapular wing of coracoid (*Clarke et al., 2006*) medial curving and further elongation of the acrocoracoid process (*Novas et al., 2021*) further reduction in angle between the scapula and coracoid (*Wellnhofer et al., 2009*) and the complete bony enclosure of triosseal canal. Regarding distinctive pectoral girdle features in particular taxa, *Jeholornis* has an unusual combination of a prominent procoracoid process and a large supracoracoid foramen (*Lefèvre et al., 2014*; *Turner et al., 2012*; *Wang et al., 2020a*), *Sapeornis* has a dorsolaterally oriented acrocoracoid process, and Enantiornithes is characterized by an extremely small scapular wing of the coracoid, a single scapula-coracoid articulation, elongation of the hypocleidium, presence of caudal grooves on the furcular rami and a keel on the caudal surface of the hypocleidium, and further solidification of the furcula-scapula articulation.

The variation in pectoral girdle morphology seen among early birds is suggestive of a similarly wide diversity of flight capabilities and modes, an inference supported by previous studies (*Close and Rayfield, 2012*; *Heers and Dial, 2012*; *Novas et al., 2021*). The position of the acrocoracoid process, or coracoid tubercle, and the orientation of the glenoid fossa are functionally important because the former is a key determinant of the course of the M. supracoracoideus tendon and the latter has a major effect on the range of motion of the wing (*Novas et al., 2020*; *Novas et al., 2021*). In volant crown birds, the tendon ascends through the triosseal canal, passes laterally over the acrocoracoid process, and ultimately inserts on the dorsal tubercle near the proximal end of the humerus. The medial surface of the acrocoracoid process forms the lateral wall of the triosseal canal and acts as a pulley to redirect the M. supracoracoideus tendon. Because the pulley is situated above the insertion point when the humerus is depressed, the force generated by the ventrally positioned belly of the M. supracoracoideus elevates the humerus, rather than protracting the humerus as in early-diverging theropods (*Burch, 2014*). In contrast to non-avialan theropods, volant crown birds are characterized by a well-developed acrocoracoid process located above the level of the glenoid fossa, which has a sub-horizontal major axis and faces laterodorsally (*Novas et al., 2020*). The M. supracoracoideus is the main elevator of the wing, and the wing moves approximately dorsoventrally at the shoulder (*Novas et al., 2020*; *Novas et al., 2021*). In *Archaeopteryx* and deinonychosaurs (e.g., *Buitreraptor* and *Sinovenator*), the hypertrophied coracoid tubercle would likewise have acted as a pulley for the M. supracoracoideus tendon (*Novas et al., 2021*), but the pulley would have been located below the level of the glenoid fossa and approximately at the dorsoventral level of the insertion point when the humerus was depressed. Thus, the M. supracoracoideus would have protracted the humerus, as in flightless extant paleognaths (*Novas et al., 2020*; *Novas et al., 2021*; *Jasinoski et al., 2006*). The glenoid fossa of *Archaeopteryx*, non-avialan pennaraptorans, and flightless paleognaths faces laterally and has a sub-vertical major axis, indicating that the movements of the forelimb at the shoulder joint are, or in the case of extinct taxa would have been, predominantly cranial-caudal (*Novas et al., 2020*; *Ostrom, 1974*).

In *Sapeornis* and most other non-ornithothoracine avialans (e.g., *Jeholornis* and *Confuciusornis*), the acrocoracoid process is slightly above the midpoint of the coracoidal glenoid fossa (*Zhou and Zhang, 2003b*), and consequently would have been above the insertion point when the humerus was depressed, as in living birds. The more dorsal location of the acrocoracoid process would have caused the tendon of M. supracoracoideus to be slightly dorsally displaced relative to its position in *Archaeopteryx* and non-avialan pennaraptorans (*Mayr et al., 2005*; *Turner et al., 2012*; *Wang and Zhou, 2018b*). The triosseal canal is located mediocranial to the glenoid fossa, whereas in extant birds the triosseal canal is located more directly medial to the glenoid fossa. In *Sapeornis,* the vector of the tension exerted by the M. supracoracoideus on the humerus would therefore have been directed cranially and somewhat dorsomedially, causing the muscle to promote protraction, elevation, and pronation of the wing during the upstroke. The angle with the vertical formed by the major axis of the

glenoid fossa is larger than in flightless paleognaths, but smaller than in volant extant birds (**Novas et al., 2020**). This suggests that the wing may have moved in a craniodorsal-caudoventral downstroke, unlike either the dorsal-ventral downstroke of extant volant birds or the largely cranial-caudal humeral movements of flightless paleognaths and presumably also of *Archaeopteryx* and non-avialan pennaraptorans. Several studies **Mayr, 2017**; **Olson and Feduccia, 1979** have suggested that the well-developed M. deltoideus, which would have inserted broadly on the deltopectoral crest and humeral shaft, played the main role in the wing elevation in *Sapeornis* and other non-ornithothoracine birds. This seems consistent with the finding in this study that the M. supracoracoideus of *Sapeornis* would have pulled the humerus cranially (**Figure 5**) rather than acting primarily as a wing elevator, and with the finding that Lig. acrocoracohumerale in *Sapeornis* had a relatively horizontal orientation, so that the dorsal shoulder musculature would have been largely responsible for preventing ventral dislocation of the humeral head when M. pectoralis was strongly activated. Nevertheless, the presence of the triosseal canal indicates that most non-ornithothoracine birds possessed some incipient capacity for powered, flapping flight. In *Piscivorenantiornis*, the acrocoracoid process is only slightly higher than the coracoidal glenoid fossa as in most non-ornithothoracine avialans, but nevertheless is considerably higher than the coracoid's scapular articular surface as in euornithines, due to the craniodorsally caudoventrally elongated shape of the coracoidal glenoid fossa. The orientation of the major axis of the glenoid fossa falls within the range seen in volant extant birds, and the triosseal canal is located medial to the glenoid fossa. Therefore, the wing movements of *Piscivorenantiornis* would have been more like those of volant extant birds than those of *Sapeornis*, indicating stronger flight capabilities in enantiornithines than in non-ornithothoracine birds.

In general, our study reveals additional lineage-specific variations in pectoral girdle anatomy as well as an overarching pattern of significant modification of the pectoral girdle along the line to crown birds. The morphological diversity seen across the pectoral girdles of Mesozoic birds presumably resulted in a commensurate range of flight capabilities and modes in early-flight evolution. The wing movements of *Sapeornis* would have differed from those of extant volant birds, highlighting the need to consider the possible effect of wing kinematics when reconstructing the flight ability of early birds.

## Materials and methods
### Institutional abbreviations
PMoL, Paleontological Museum of Liaoning, Shenyang, China; IVPP, Institute of Vertebrate Paleontology and Paleoanthropology, Chinese Academy of Sciences (CAS), Beijing, China; STM, Shandong Tianyu Museum of Natural History, Linyi, China.

*S. chaoyangensis* PMoL-AB00015 is a nearly complete semi-articulated skeleton collected from the Lower Cretaceous Jiufotang Formation at Yuanjiawa Village, Dapingfang Town, Chaoyang County, Liaoning Province, China. It is probably an adult individual based on skeletal fusion features (e.g., closed neurocentral sutures in all vertebrae, sacral vertebrae fused to form a synsacrum, distal carpals fused with metacarpals to form a carpometacarpus, proximal tarsals fused with tibia to form a tibiotarsus, and distal tarsals fused with metatarsals to form a tarsometatarsus). *P. inusitatus* IVPP V 22582 is a disarticulated skeleton collected from the Jiufotang Formation near Dapingfang Town (*Wang and Zhou, 2017Wang and Zhou, 2017*; *Wang et al., 2016a*). The skeleton shows the same fusion features as PMoL-AB00015 and probably also represents an adult individual. These two specimens are the main data sources for this study. For comparison, we also examined the skeletons of several extant birds and fossil theropods, including *T. alba* IVPP OV 954, *E. garzetta* IVPP OV 1631, *P. muticus* IVPP OV 1668, and the troodontid *Sinovenator changii* IVPP V 12615. The specimens were scanned using a GE v|tome|x m300&180 micro-computed tomography scanner (GE Measurement & Control, Wuntsdorf, Germany) and a 225 kV micro-computerized tomography scanner (developed by the Institute of High Energy Physics, CAS), both housed at the Key Laboratory of Vertebrate Evolution and Human Origins of the Chinese Academy of Sciences. Three-dimensional segmentation of the computed tomography data was performed using the software package Mimics (19.0).

## Acknowledgements
We thank Xiaoqing Ding for specimen preparation; Minghui Ren for illustration; Yun Feng and Yemao Hou for help with computed tomography scanning; and Paul Rummy for discussion. We thank the

reviewers Jingmai K O'Connor and Fernando E Novas for their helpful and constructive comments that greatly contributed to improving this manuscript. This work was supported by the National Natural Science Foundation of China (41688103, 42288201, 42072030), the International Partnership Program of Chinese Academy of Sciences (132311KYSB20180016), Natural Sciences and Engineering Research Council of Canada funding (Discovery Grant RGPIN-2017-06246), and start-up funding awarded by the University of Alberta to CS.

## Additional information

### Funding

| Funder | Grant reference number | Author |
| --- | --- | --- |
| National Natural Science Foundation of China | 41688103 | Xing Xu<br>Dongyu Hu |
| National Natural Science Foundation of China | 42288201 | Xing Xu<br>Dongyu Hu |
| National Natural Science Foundation of China | 42072030 | Xing Xu<br>Dongyu Hu |
| International Partnership Program of Chinese Academy of Sciences | 132311KYSB20180016 | Xing Xu |
| Natural Sciences and Engineering Research Council of Canada | Discovery Grant RGPIN-2017-06246 | Corwin Sullivan |
| University of Alberta | | Corwin Sullivan |

The funders had no role in study design, data collection and interpretation, or the decision to submit the work for publication.

### Author contributions

Shiying Wang, Conceptualization, Visualization, Writing – original draft, Writing – review and editing; Yubo Ma, Qian Wu, Min Wang, Corwin Sullivan, Writing – review and editing; Dongyu Hu, Conceptualization, Writing – review and editing; Xing Xu, Conceptualization, Writing – original draft, Writing – review and editing

### Author ORCIDs

Shiying Wang ⓘ http://orcid.org/0000-0001-6067-0303
Min Wang ⓘ http://orcid.org/0000-0001-8506-1213
Xing Xu ⓘ http://orcid.org/0000-0002-4786-9948

### Decision letter and Author response

Decision letter https://doi.org/10.7554/eLife.76086.sa1
Author response https://doi.org/10.7554/eLife.76086.sa2

## Additional files

### Supplementary files

• Transparent reporting form

### Data availability

All data are available in the article.

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
