## [Editor Report]

The authors provide new 3D fossil findings in *Sapeornis*, an avialan that lived during the Early Cretaceous period, a key node in our understanding of the evolution of avian flight. The functional reconstruction of two critical skeletal elements of the avian flight apparatus, the scapula and coracoid, enables the authors to hypothesize how the evolution of the scapula and coracoid enabled the modern avian wing stroke. The new 3D morphological reconstruction enables future integrative studies of *Sapeornis* flight performance based on biomechanical, muscle physiological, and aerodynamic principles and is thus a key building block to inform our general understanding of the evolution of avian flight.

---

## [Decision Letter]

**Decision letter after peer review:**

Thank you for submitting your article "Digital restoration of the pectoral girdles of two Early Cretaceous birds, and implications for early flight evolution" for consideration by *eLife*. Your article has been reviewed by 2 peer reviewers, and the evaluation has been overseen by a Reviewing Editor and George Perry as the Senior Editor. The following individuals involved in review of your submission have agreed to reveal their identity: Jingmai O'Connor (Reviewer #1); Fernando E. Novas (Reviewer #2).

The reviewers have discussed their reviews with one another, and the Reviewing Editor has drafted this to help you prepare a revised submission. Please comply with the comments and suggestions to your best ability, mark all changes using a blue font to indicate changes in the revised manuscript, and respond in a point-by-point fashion. This is essential to enable the reviewing editor to fully evaluate the merit of your revision.

Essential revisions:

Overall, both the reviewers and the editors are excited by the quality of the research, the following weaknesses and comments remain to be addressed.

General Comments:

1. Overall, the presentation of the 3D morphological data in the figures is not sufficiently easy to interpret for the general *eLife* reader and as such this manuscript is focust on specialist only in its current format. To make the manuscript acceptable for *eLife* the following changes are needed. (i) There is a strong need for an introductory figure introducing all key morphological concepts in the context of the entire inferred body plan of the animals studied. This introductory figure should also show the assumed behavioral – locomotion – context that the work used to interpret the functional anatomy of the fossils. (ii) All the current figures have specialist labeling, the labeling and figures require avatars showing the location of the bones in the body plan with common wording and labels any *eLife* reader can understand. Only once the labels have been introduced with common anatomical nomenclature, can shorthand names be used when absolutely essential for the rigor of the science. Whenever common nomenclature is sufficient, that is preferred so it is easier for the general *eLife* reader to interpret and understand the findings presented.

2. It is unclear if the PMOL specimen is a juvenile or subadult? >> Please avoid using acronyms in the manuscript, *eLife* provides as much space as the authors need to not use acronyms and make the manuscript more accessible to *eLife*'s entire readership. << Discuss the matter of fusion between the scapulocoracoid in Sapeornis and how that may (or may not) affect the musculature. This is mentioned for other taxa in the discussion but apparently not for Sapeornis.

3. Please expand the description to include the detail that a general *eLife* reader needs to comprehend the morphology. The current descriptions are too vague. For example, the manuscript states the scapula is twisted, but information on how it is twisted is missing in the text and the figures – clear avatars would help the general reader see this in the figures. Another example "A broad flange projects costally…" It is unclear what the shape of the flange is, how far it projects, how long it is, etc. This lack of quantitative information and description makes the paper too subjective for the *eLife* readership to follow and it makes it hard for specialists to fully appreciate the reported findings and integrate it in their future research.

4. The division between the scapular and sternal portions of the coracoid are currently confusing and give the impression it is almost arbitrary, especially because the sternal portion in birds still articulates with the scapula in some taxa in figure 3, e.g. enantiornithines. If this way of describing the two parts of the plesiomorphic coracoid is commonly used in the specialist literature, please keep in mind this manuscript has been submitted to *eLife*, which serves an extraordinary broad audience. So in addition to clarifying these matters in the figures and text, please add citations so the *eLife* readership can also find the pertinent specialist context in the literature that may motivate some of the authors choices. However, in case it is new, we don't recommend moving forward with this way of describing, or rather dividing, the coracoid and instead serve the broad readership. One of our technicalconcerns is that if this section houses the supracoracoidal nerve foramen, then it doesn't quite work when the SCNF is in the neck of the coracoid in enantiornithines (see the Buffetaut 1998 specimen and Lecho specimens) and some modern birds. It would be more helpful to either mark the glenoid and acrocoracoid etc. in a distinct color instead of coloring the two supposed parts of the coracoid. This would better illustrate the revised text.

5. It is necessary to be more consistent in using either anterior/cranial and posterior/caudal to serve *eLife*'s readership, please also introduce any differences between your field's preferred choices and common choices in other biological fields so the manuscript better serves all readers. Please also visually introduce the chosen definitions in figure one in the context of the body plan so any reader can follow and integrate your contribution crossdisciplinary. Similarly, in Sapeornis the two major surfaces of the coracoid are described as dorsal and ventral but it changes to cranial and caudal in Piscivorenantiornis despite the fact the orientation of the coracoid doesn't change according to Figure. 4 (and also Baumel uses dorsal and ventral for these two surfaces). Consistency in directional terms will improve the readability of this manuscript although for non-avian theropods different terms are used, we suggest using all avian terms and putting the theropod directional language in parentheses when talking about taxa like Sinovenator and illustrating these different perspectives in the new introductory figure 1 which will probably have to be a multipaneled figure to serve *eLife*'s readership). Please also add a short section in the beginning in which the directional terminology used throughout is explained explicitly in the text with clear references to the visual illustration in the new figure 1.

6. To further help *eLife*'s readership and specialist alike, the new figure 1 should also include the following. Comparative images of scapular and coracoid shape in more taxa would, even if they are not based on digital data, which is why we recommend adding these in the new introductory figure 1. E.g. Archaeopteryx, Jeholornis, Confuciusornis are all mentioned but there are no images to help orient the reader and interpret the comparisons that are made due to missing critical visual information that is assumed known – which is not the case for *eLife*'s readership since the journal serves the entire biological (neuroscience, medical as well as several other disciplines including biophysics, biomathematics, bioinspired engineering etc.). This should also be paired with indications of the range of motion that are mentioned in the discussion for greater clarity. The discussion should also mention the limitations in how these ranges of motion have previously been established and how this limits the current analysis in a general biomechanical functional framework. Additionally, visually indicating the attachment of major muscles discussed in the new introductory figure 1, in the context of the body plan, would also be very helpful to orient the reader and follow the discussion. To help the reader further, the introduction, results and Discussion sections should reference to this introductory figure whenever a new concept is discussed in the text (please don't assume the readers find these connections obvious). *eLife* does not have length restrictions for papers, so the authors can invest words and figures to serve the readers.

7. There seems to be no acknowledged that the morphology of the enantiornithine pectoral girdle in 3D already known from Late Cretaceous specimens, but that the new morphology presentation is clearer for Early Cretaceous specimens, and it can now be recognized that many Late Cretaceous morphologies are also present in Early Cretaceous specimens, which previously could not be recognized without 3D CT data. Please provide this context in the introduction and discussion.

8. It would be helpful to discuss the homoplasy affecting the evolution of these two bones using non specialist wording, so the general implications are clearer to all readers. Additionally, it would be great to discuss how the absence of an ossified sternum would affect the flight musculature in Sapeornis and what further research could provide more decisive evidence.

9. In the discussion, when all the morphological transitions are discussed, citations should be provided for where these have been discussed previously. And where possible, please illustrate them in the new introductory figure 1 so all readers can comprehend the significance of the current research in the literature context.

Detailed Comments:

1. In figure 2 the Fura abbreviation is missing from caption; also indicate where the furcula contacts the scapula in B and D. Ideally as few as possible abbreviations are used in all figures, it would better serve the reader to use the figure space that *eLife* provides to use the full names and make the figures highly readable to all. In particular regarding the key morphological aspects researched and discussed. It is understandable that the authors may choose to abbreviate aspects that are only indicated for the specialist reader and are not essential to interpret and understand the work, however one could question if indicating non-essential information is even needed. So, abbreviations should be used sparingly with a bias towards making all labels readable for all – without abbreviation.

2. Please add the coracoclavicular ligament in Figure 4 add to the other models (currently only Sapeornis) and show the triosseal canal in lateral view as well (or rather the path of the supracoracoideus ligament). Because it is very hard to see the location of the triosseal canal in the figure as is, because the illustrations are proportionately small. Again *eLife* provides all figure space needed to make the figures readable to both specialist (who could not read it) and the rest of *eLife*'s readership (who need even clearer figures).

3. Line 29 – The statement that some taxa have "one area" and others a "double articulation" is confusing. Please specify which is "the double articulation" mentioned as widely present among pennaraptorans? Please provide all context (visual, text, references) needed to follow the point.

4. Line 32 – Please resolve the readers confusion regarding the sharp cut in the descriptive process. E.g. the treatment of the presence of triosseal canal among birds, without mentioning the (eventual) relation with the "only one vs double articulation" condition analyzed before.

5. Line 33 – Is this "transitional stage" first recognized here? What are the nodes representing the transition? Please provide all context (visual, text, references) needed to follow the point.

6. Line 34-37 – Please explain the importance of "only one vs double scap-cor articulation" and the "transitional stage of triosseal canal" in the context of the anatomical modifications that occurred in the line to birds. How do the new observations on these two aspects modify and expand previous hypotheses on the origin and evolution of bird flight? Please provide all context (visual, text, references) needed to follow the point.

7. Line 44 – Consider citing Ostrom 1976 for his role in this line of research.

8. Line 50-51 – Please, consider the paper by Imai et al., 2019 on Fukuipteryx. May these authors already have presented a 3D reconstruction of the scapular girdle of the kind present here for other taxa? Being first is not considered important for an *eLife* publication, as a true first rarely exist in science, we are all building off the research by others and this work should be clearly cited and discussed so that science robustly advances.

9. Line 74-75 – There may be some context missing regarding the diversity in ideas of how dromaeosaurid relates to Rahonavis. Novas et al., (2018. "Postcranial osteology of a new specimen of Buitreraptor". Cretaceous Research) discuss features that Rahonavis shares with birds more derived than dromaeosaurids and Archaeopteryx (particularly the size and shape of the acromial process). This work suggests that Rahonavis is a long-tailed avialan. Whereas the new work seems to support support the idea that the taxon is not a dromaeosaurid. Discussing how the new work may enrich or revise previous perspectives with citations will help the general reader better understand the contribution made in the context of the literature. Please revise the text in a fashion that you consider most rigorous and aligned with the new fossil insights.

10. Line 80 – Please clarify if this may be related with the ventral surface of the acromial process.

11. Line 81 – It may help the reader to also indicate the position with respect to the glenoid. E.g. aside from being "more medially positioned", is it anteriorly/posteriorly/at level with the glenoid? Please provide all context (visual, text, references) needed to follow the point.

12. Line 111 – Consider revising this as follows: "…the acrocoracoid process (frequently described as biceps or coracoid tubercle)". Considering Archaeopteryx is an avialan sharing a common ancestor with Sapeornis and the rest of the birds, for which the term "acrocoracoid" is used. Afterall is a matter of nomenclature, and there is agreement that such tubercle is homologous. Please provide context in the text using references to other readers outside the field can follow.

13. Line 113 – Was this morphology confirmed in other specimens of Sapeornis? Is it natural or is it the result of postmortem compression? Please provide all context (visual, text, references) needed to follow the point, considering not all *eLife* readers are familiar with how postmortem compression may typically be visible in fossils.

14. Line 114 – Please, check the notes regarding Figure 1G,H for correctness and clarity.

15. Line 115-118 – Such a fossa on the acrocoracoid process is also seen in Buitreraptor and Rhea (colloquially it is a little like a volcano with a crater). Novas et al., (2021) interpreted this fossa as the site of attachment of the acrocoraco-acromial ligament. The acrocoracoacromial ligament forms a bridge under which the m. supracoracoideus slides. This ligament is also present in modern flying birds, but it is cranially covered by the omal end of the furcula. Please better integrate the literature context in alignment with the perspective that the new data provides and clearly delineate quantitative evidence versus qualitative interpretation and any further work that may be needed to rigorously conclude.

16. Line 131 – Please provide an extra figure, a clear line drawing that relates back to the new introductory figure 1, depicting the correspondence of the articular areas? This is important for other authors studyingh these features in other paravians. Please provide all context (visual, text, references) so other workers can build off your research.

17. Line 140-141- Based on Figure 4 (Sapeornis), the triosseal canal width, could the tendon of the m. supracoracoideus have been substantially thicker than in other avialans, considering the reconstructed diameter of the canal? Please provide all context (visual, text, references) so other workers can build off your research.

18. Line 154-155 – Please condiseder and integrate in your discussion that Rahonavis may be a long-tailed avialan, instead of a deinonychosaur. To support your perspective based on the presented data, please provide all context (visual, text, references) so the readers can follow your interpretations and comprehend any limitations and open discussion points so science advances robustly.

19. Line 206- Please provide the species name so the general reader can follow.

20. Line 237- Considering Rahonavis exhibits derived features shared with birds more derived than Archaeopteryx, which seem absent in deinonychosaurs and unenlagiines, the manuscript could evaluate the scapular features of Rahonavis, to provide more concludive support to either deinonychosaur or avialan affinities of this taxon. Please provide this in a context that aligns best with the new 3D fossil information and functional morphological analyses.

21. Line 273-275- The fossils presented support this condition. We wonder, however, if enantiornithes may have had (in life) a procoracoidal section (cartilaginous) for contacting with the well-developed acromial region of the proximal scapula. A "proto-procoracoid" or "supracoracoid canal" seems to have been present and well developed in Buitreraptor, and presumably served as a "channeling surface" for the m. supracoracoid towards the proximal end of humerus. Crown birds seem to have a procoracoid, and a similar channeling surface is thought to be present among basal paravians. We thus wonder if Enantiornithes may have had such structure, but in an unossified form. Please consider these options and clarify the manuscript to better substantiate your interpretation of the new 3D fossil evidence.

22. Line 290-300 – There is no need to repeat this since it was clearly explained in previous lines.

23. Line 313-315 – is "roofed" the correct wording? May the procoracoid be forming the caudomedial wall of the canal, instead of a roof? Please help the general reader correctly interpret the fossil evidence and clarify the text and figure labeling and orientation of the reader to fully resolve this.

24. Line 322 – Please consider that in Rhea the acrocoraco-acromial ligament bridges the tendon of m. supracoideus, enabling it to functionally act as a "triosseal canal" as discussed by Novas et al., 2021.

25. Line 338-340 – The following is confusing "in basal paravians as well as in living birds the m. supracoracoid runs along the LATERAL side of the acromion" for the reader. The medial surface of the later one is contiguous with the medial surface of scapular blade, thus the medial surface of the acromion contacts with the underlying thorax. Hence it seems more logical to consider the supracoracoid doesn´t run between the scapula and ribs. Please reconsider the wording and clarify the text (as needed with a figure and references) so the general reader can follow the authors reasoning or correct the error.

26. Line 348-349 – The presence of a triosseal canal is defined by the contact between the acrocoracoid and epicleidium. If such a contact does not exist, then such a "triosseal" canal is not defined. Hence this is also a semantic problem, please resolve so a general reader can comprehend.

27. Regarding the function of the scapular girdle in paravians: The deflection of the tendon may have been present in basal paravians, and the "inner half" of the triosseal canal seems to have already been present in basal paravians. The shape of the coracoid of Sapeornis (the development and position of acrocoracoid) seems to resemble that of Archaeopteryx and Buitreraptor, hence it may be reasonable to assume the "pulley system" may have functioned in a similar way in all these forms. Hence one could consider the term "triosseal canal" to be inadequate for Sapeornis. Further, a similar pulley-shaped morphology for coracoid was also reported for basal paravians (Buitreraptor, Archaopteryx; Novas et al., 2021). Please discuss how the new 3D fossil evidence may or may not support this line of thought, in case you end up agreeing, please revise the text accordingly and otherwise please clarify the text and cite supporting literature so the reader can follow the authors reasoning.

28. Line 375-376 – Figure 4 seems to suggest that Sapeornis does exhibit a derived acrocoracoid, because it is positioned at level with the glenoid, and not in front of it. Hence it is not fully clear why the "extension of acrocoracoid process dorsal to level of coracoidal glenoid fossa" (a condition that seems absent in Sapeornis) is interpreted as characteristic of Step I. In case you end up agreeing, please revise the text accordingly and otherwise please clarify the figure and text and cite supporting literature so the reader can follow the authors reasoning.

29. Line 403-407 – Novas et al., (2020, 2021) suggest that the pulley was already operative in basal paravians such as Buitreraptor and Archaeopteryx, but the m. supracoracoideus acted as a humeral protractor (as in living ratites). This condition is intermediate between the humeral depressor action (in basal theropods) and humeral elevator role (in extanct birds). Novas et al., (2020, 2021) did not state that the "m. supracoracoideus would have acted straightforwardly as a wing depressor". Please resolve.

30. Line 419-422 – The beautifully illustration of the coracoid of Sapeornis does not show a condition of the acrocoracoid that is clearly different from Archaeopteryx. The acrocoracoid process does not seem to be in front of the glenoid, but almost at the same level. So, could it be more reasonable to consider the m. supracoracoideus in Sapeornis to have functioned as a humeral protractor, not as an elevator? Please resolve.

31. Line 720 (including the labels in Figure 1H) – Is the border closer to the sternum? If so, it seems "distal" may be more appropriate in the text (considering the glenoid represents the "proximal end" of the coracoid). Please clarify and resolve so the general reader can follow the authors reasoning in the context of the literature.

*Reviewer #1 Recommendations for the authors:*

Great manuscript, it is really exciting to see more 3D data becoming available for important taxa like Sapeornis and for the morphology of the musculature in these early birds to begin to be discussed (and also reconstructed, see comment below – it is a bit difficult to visualize this information for someone who is not a myologist like myself).

Is the PMOL specimen a juvenile or subadult? Discuss the matter of fusion between the scapulocoracoid in Sapeornis and how that would affect the musculature (if at all). This is mentioned for other taxa in the discussion but not for Sapeornis itself

I would suggest the description include greater detail to clarify the morphology. As written the description is a little vague (examples, the scapula is twisted, twisted how? A broad flange projects costally…. what is the shape of the flange? how far does it project, how long is it, etc.).

I find the division between the scapular and sternal portions of the coracoid confusing and almost arbitrary, especially because the sternal portion in birds still articulates with the scapula in some taxa in figure 3 (including enantiornithines). If this way of describing the two parts of the plesiomorphic coracoid is commonly used in the literature and I am simply unaware of this, then please add citations. However, if this is new, I would not recommend moving forward with this way of describing or rather dividing the coracoid. I understand that it seems superficially that this is the portion of the coracoid that becomes reduced but thats not quite accurate. If this section houses the supracoracoidal nerve foramen then it doesn't quite work when the SCNF is in the neck of the coracoid in enantiornithines (see the Buffetaut 1998 specimen and Lecho specimens) and some modern birds. It would be more helpful to either mark the glenoid and acrocoracoid etc in a distinct color (rather than color the two supposed parts of the coracoid). This would better supplement the text.

It is necessary to be more consistent in using either anterior/cranial and posterior/caudal (choose one). Similarly in Sapeornis the two major surfaces of the coracoid are described as dorsal and ventral but it changes to cranial and caudal in Piscivorenantiornis despite the fact the orientation of the coracoid doesn't change according to Figure. 4 (and also Baumel uses dorsal and ventral for these two surfaces). Consistency in directional terms will improve the readability of this manuscript (although for non-avian theropods different terms are used, I would suggest using all avian terms and putting the theropod directional language in parentheses when talking about taxa like Sinovenator). I think it is worth adding a short section in the beginning in which the directional terminology used throughout is explained explicitly.

Comparative images of scapular and coracoid shape in more taxa would be helpful, even if they are not based on digital data. Archaeopteryx, Jeholornis, Confuciusornis are all mentioned but there are no images to help the reader interpret the comparisons that are made. This could also be paired with indications of the range of motion that are mentioned in the discussion for greater clarity. Additionally, indicating the attachment of major muscles discussed would also be very helpful.

It should be acknowledged that the morphology of the enantiornithine pectoral girdle in 3D was well known from Late Cretaceous specimens but that now the morphology is clearer for Early Cretaceous specimens and it can now be recognized that many Late Cretaceous morphologies are also present in Early Cretaceous specimens but previously could not be recognized without 3D CT data.

It would be helpful to discuss the homoplasy affecting the evolution of these two bones. Additionally, it would be great to discuss how the absence of an ossified sternum would affect the flight musculature in Sapeornis.

In the discussion, when all the morphological transitions are discussed, citations should be provided for where these have been discussed previously.

Figure 2 Fura abbreviation is missing from caption; also indicate where the furcula contacts the scapula in B and D.

In Figure 4 add the coracoclavicular ligament to the other models (currently only Sapeornis) and show the triosseal canal in lateral view as well (or rather the path of the supracoracoideus ligament); it is very hard to see the location of the triosseal canal in the figure as is because the illustrations are proportionately small.

*Reviewer #2 Recommendations for the authors:*

Line 29 – I am confused with the statement that some taxa have "one area" and others a "double articulation". Could you specify which is "the double articulation" mentioned as widely present among pennaraptorans?

Line 32 – Here produces a sharp cut in the descriptive process, with the treatment of the presence of triosseal canal among birds, without mentioning the (eventual) relation with the "only one vs double articulation" condition analyzed before.

Line 33 – Is this "transitional stage" firstly recognized here? Could you precise which are the nodes representing such transition?

Line 34-37 – I agree with the authors in the conclusions they express in this paragraph. However, I strongly suggest to explain the importance of the "only one vs double scap-cor articulation" and the "transitional stage of triosseal canal" in the anatomical modifications occurred in the line to birds. How their observations on these two aspects modify and expand previous hypotheses on the origin and evolution of bird flight?

Line 44 – Let me suggest to cite here Ostrom 1976, which I believe is the founder of this line of studies

Line 50-51 – Please, check the paper by Imai et al., 2019 on Fukuipteryx. Do these authors already present a 3D reconstruction of the scapular girdle of the kind you present here for other taxa? Anyway, to be the first or not is secondary, and present manuscript represents a formidable progress in our knowledge on the flight apparatus of early birds

Line 67 – These warnings are necessary and welcome. Many papers on scapulocoracoid in paravian theropods overlook describing the state of preservation of the available materials.

Line 74-75 – I am respectful of the interpretation that current authors are following about the dromaeosaurid affiliation of Rahonavis. However, let me say that Novas et al., (2018. "Postcranial osteology of a new specimen of Buitreraptor". Cretaceous Research) emphasized about the presence of many features that Rahonavis shares with birds more derived than dromaeosaurids and Archaeopteryx (particularly the size and shape of the acromial process!), suggesting that Rahonavis is, in fact, a long-tailed avialan. Interestingly, you are here noting on a quite derived morphology of the acromion, being present in Jeholornis and Rahonavis, thus lending support to the idea that the later taxon is not a dromaeosaurid.

Line 80 – Is it related with the ventral surface of the acromial process?

Line 81 – Let me suggest to indicate also the position with respect to the glenoid. Aside from being "more medially possitioned", is it anteriorly/posteriorly/at level with the glenoid?

Line 111 – Let me suggest to change this part of the phrase, saying something like the following: "…the acrocoracoid process (frequently described as biceps or coracoid tubercle)". The reason is that Archaeopteryx is an avialan sharing a common ancestor with Sapeornis and the rest of the birds, for which the term "acrocoracoid" is used. Afterall is a matter of nomenclature, and all of us agree that such tubercle is homologous.

Line 113 – Did you check such morphology in other specimens of Sapeornis? Is it natural or is it the result of postmortem compression?

Line 114- Please, check some notes on Figure 1G,H

Line 115-118 – Such a fossa on the acrocoracoid process is also seen in Buitreraptor and Rhea, for example (it is something like a volcano with a crater on its top). Novas et al., (2021) interpreted this fossa as the site of attachment of the acrocoraco-acromial ligament. The acrocoracoacromial ligament forms a bridge under which the m. supracoracoideus slides. This ligament is also present in modern flying birds, but it is cranially covered by the omal end of the furcula.

Thus, keep in mind this in your considerations that the fossa on the acrocoracoid process served for attachment point for a coracoclavicular ligament connecting the coracoid and furcula (an interpretation that I am not dismissing).

Line 131 – Could you provide an extra figure (a simple line drawing) depicting the correspondence of articular areas? This could be important for other authors in searching for these features in other paravians.

Line 140-141- This is an interesting interpretation. By observing Figure 4 (Sapeornis), the triosseal canal results wide. I wonder if the tendon of the m. supracoracoideus was very thick, or thicker than in other avialans, based on the reconstructed diameter of the canal.

Line 154-155 – keep in mind that Rahonavis may be a long-tailed avialan, instead of a deinonychosaur.

Line 206- Please, insert species name.

Line 237- Let me express again that Rahonavis exhibits many derived features shared with birds more derived than Archaeopteryx, which are absent in deinonychosaurs and unenlagiines. Probably the present manuscript may represents a good place to evaluate the scapular features of Rahonavis, lending support to either deinonychosaur or avialan affinities of this taxon.

Line 273-275- I agree, of course, with this conclusion. Fossil bones are clear in indicating this condition. I wonder, however, if enantiornithes had (in life) a procoracoidal portion (cartilaginous) for contacting with the well developed acromial region of proximal scapula. A "proto-procoracoid" or more properly a "supracoracoid canal" was present and well developed in Buitreraptor, and presumably served as a "channeling surface" for the m. supracoracoid in its course towards the proximal end of humerus. Crown birds have procoracoid, and a similar channeling surface was present among basal paravians. Enantiornithes may have had such structure, but in an unossified condition.

Line 290-300- I believe it is unnecessary to repeat. Authors have already clearly explained this in previous lines.

Line 313-315 – s it right to say "roofed"? Is the procoracoid forming the caudoMEDIAL wall of the canal, instead of a roof? Apologies if I am wrong.

Line 322- In Rhea the acrocoraco-acromial ligament bridges the tendon of m. supracoideus, functionally acting as a "triosseal canal". See Novas et al., 2021 for discussions on these asepcts.

Line 338-340- I am surprised for this statement: in basal paravians as well as in living birds the m. supracoracoid runs along the LATERAL side of the acromion. The medial surface of the later one is contiguous with the medial surface of scapular blade, thus the medial surface of the acromion contacts with the underlying thorax. The supracoracoid doesn´t run between the scapula and ribs!

Line 348-349- The presence of a triosseal canal is defined by the contact between the acrocoracoid and epicleidium. If such a contact does not exist, then such a "triosseal" canal is not defined. of course, this is a semantic problem.

A completely different matter is the function that this region of the scapular girdle played in paravians: a well defined though is present on the craniomedial surface of coracoid in Buitreraptor which may have act as a "proto-procoracoid" for chanalizing the m. scapulocoracoid. Thus, the deflection of its tendon was probably operative in basal paravians, and the "inner half" of the triosseal canal was already present even in basal paravians. The shape of the coracoid of Sapeornis (especially the development and position of acrocoracoid) resembles that of Archaeopteryx and Buitreraptor, thus it is expectable that the "pulley system" was postioned and functioned in a similar way in all these forms.

In sum, I agree with authors regarding the function of this region of scapular girdle, but the term applied ("triosseal canal") for Sapeornis seems inadequate. Besides, but no less important, a similar pulley-shaped morphology for coracoid was also identified for basal paravians (Buitreraptor, Archaopteryx; Novas et al., 2021).

Line 375-376- Notably, as you shown in Figure 4, Sapeornis does not exhibit a derived condition of acrocoracoid, because it is at level with the glenoid, not in front of it. Thus, I don´t understand why the "extension of acrocoracoid process dorsal to level of coracoidal glenoid fossa" (a condition that seems absent in Sapeornis), is interpreted as characteristic of Step I.

Line 403-407- Novas et al., (2020, 2021) said that the pulley was already operative in basal paravians such as Buitreraptor and Archaeopteryx, but the m. supracoracoideus acted as a humeral PROTRACTOR (as in living ratites). This condition is intermediate between the humeral depressor action (in basal theropods) and humeral elevator role (in extanct birds).

Novas et al., (2020, 2021) did not say that the "m. supracoracoideus would have acted straightforwardly as a wing depressor".

Line 419-422- Based on what is beautifully illustrated in present manuscript, the coracoid of Sapeornis does not show a condition of the acrocoracoid sharply different from that in Archaeopteryx. In other terms, the acrocoracoid process is not in front of the glenoid, but almost at the same level. So, I will say the m. supracoracoideus in Sapeornis functioned as a humeral protractor, not as an elevator.

Line 720 (and labels inserted in Figure 1H)- Is this the border closer to the sternum? Then, it has to be "distal" using the terminology applied on the text (the glenoid represents the "proximal end" of the coracoid).

---

## [Author Response]

General Comments:1. Overall, the presentation of the 3D morphological data in the figures is not sufficiently easy to interpret for the general eLife reader and as such this manuscript is focust on specialist only in its current format. To make the manuscript acceptable for eLife the following changes are needed. (i) There is a strong need for an introductory figure introducing all key morphological concepts in the context of the entire inferred body plan of the animals studied. This introductory figure should also show the assumed behavioral – locomotion – context that the work used to interpret the functional anatomy of the fossils. (ii) All the current figures have specialist labeling, the labeling and figures require avatars showing the location of the bones in the body plan with common wording and labels any eLife reader can understand. Only once the labels have been introduced with common anatomical nomenclature, can shorthand names be used when absolutely essential for the rigor of the science. Whenever common nomenclature is sufficient, that is preferred so it is easier for the general eLife reader to interpret and understand the findings presented.

Following the reviewers’ suggestion, we provided an introductory figure to illustrate the key morphological concepts, and we even added a paragraph to introduce the general context; we also added cartoon illustration to show the anatomical position of the structures that we described in the manuscript; we deleted nearly all shorthand names and replace them with the common nomenclature. In doing this, we are able to help the general *eLife* readers to understand the manuscript more easily as the editor suggested.

2. It is unclear if the PMOL specimen is a juvenile or subadult? >> Please avoid using acronyms in the manuscript, eLife provides as much space as the authors need to not use acronyms and make the manuscript more accessible to eLife's entire readership. << Discuss the matter of fusion between the scapulocoracoid in Sapeornis and how that may (or may not) affect the musculature. This is mentioned for other taxa in the discussion but apparently not for Sapeornis.

Following the reviewers’ suggestion, we added information on the ontogenetic stage of both PMoL-AB00015 and IVPP V 22582. Both specimens are probably adult individuals based on skeletal fusion features.

Following the reviewers’ suggestion, we have taken out all acronyms from the manuscript.

Following the reviewers’ suggestion, we added data on whether *Sapeornis* has fused scapulocoracoid. When the M. supracoracoideus contracts, the triosseal canal supports the tendon in almost the same way whether coracoid fused with the scapula or not. Thus, we do not think the fusion state would strongly affect the musculature.

3. Please expand the description to include the detail that a general eLife reader needs to comprehend the morphology. The current descriptions are too vague. For example, the manuscript states the scapula is twisted, but information on how it is twisted is missing in the text and the figures – clear avatars would help the general reader see this in the figures. Another example "A broad flange projects costally…" It is unclear what the shape of the flange is, how far it projects, how long it is, etc. This lack of quantitative information and description makes the paper too subjective for the eLife readership to follow and it makes it hard for specialists to fully appreciate the reported findings and integrate it in their future research.

Following the reviewers’ suggestion, we added new figures and new text to help the readers understand more easily the morphology. However, in some cases, we are unable to provide details (e.g., the flange is badly preserved, and we are not able to tell its exact shape and size)

4. The division between the scapular and sternal portions of the coracoid are currently confusing and give the impression it is almost arbitrary, especially because the sternal portion in birds still articulates with the scapula in some taxa in figure 3, e.g. enantiornithines. If this way of describing the two parts of the plesiomorphic coracoid is commonly used in the specialist literature, please keep in mind this manuscript has been submitted to eLife, which serves an extraordinary broad audience. So in addition to clarifying these matters in the figures and text, please add citations so the eLife readership can also find the pertinent specialist context in the literature that may motivate some of the authors choices. However, in case it is new, we don't recommend moving forward with this way of describing, or rather dividing, the coracoid and instead serve the broad readership. One of our technicalconcerns is that if this section houses the supracoracoidal nerve foramen, then it doesn't quite work when the SCNF is in the neck of the coracoid in enantiornithines (see the Buffetaut 1998 specimen and Lecho specimens) and some modern birds. It would be more helpful to either mark the glenoid and acrocoracoid etc. in a distinct color instead of coloring the two supposed parts of the coracoid. This would better illustrate the revised text.

Following the reviewers’ suggestion, we explicitly defined the scapular and sternal wings of the coracoid in the text and we also illustrated the two wings in the newly added figure. Such a division was originally proposed by Xu (2002) in which the author suggests a major modification in coracoid evolution is the appearance of biplanar morphology. More specifically, in ornithomomiosaurs and other maniraptoriforms, the distal portion of the coracoid (called distal ramus in Xu 2002, but sternal wing in this manuscript because ramus normally refers to an elongated structure) is deflected from the proximal ramus (the scapular wing in this manuscript), and this distal ramus (the sternal wing) becomes larger in size and with a nearly straight edge for articulating the sternum in late-diverging maniraptoriforms such as the dromaeosaurid *Sinornithosaurus* (see the following figure). In some species, there is a ridge emanating from the coracoid tubercle to delimit the boundary between the scapular and sternal wings (see the following figure), and in other species, a distinct ridge is absent, but a deflecting zone is present. It is the sternal wing that turns into the main body of the strut-like coracoid in most avialans; the scapular wing is highly reduced, with its sheet-like medial portion turning into the procoracoid process in many birds and the lateral portion (the main portion for articulating the scapula) shortened and moving to the base of the acrocoracoid process. The supracoracoid foramen is present within the scapular wing in most species, though in species that has a highly reduced scapular wing, the supracoracoid foramen is present in the sternal wing (in some species this foramen is absent). Nevertheless, following the reviewers’ suggestion, we also colored some other important parts of the coracoid such as the glenoid and acrocoracoid.

5. It is necessary to be more consistent in using either anterior/cranial and posterior/caudal to serve eLife's readership, please also introduce any differences between your field's preferred choices and common choices in other biological fields so the manuscript better serves all readers. Please also visually introduce the chosen definitions in figure one in the context of the body plan so any reader can follow and integrate your contribution crossdisciplinary. Similarly, in Sapeornis the two major surfaces of the coracoid are described as dorsal and ventral but it changes to cranial and caudal in Piscivorenantiornis despite the fact the orientation of the coracoid doesn't change according to Figure. 4 (and also Baumel uses dorsal and ventral for these two surfaces). Consistency in directional terms will improve the readability of this manuscript (although for non-avian theropods different terms are used, we suggest using all avian terms and putting the theropod directional language in parentheses when talking about taxa like Sinovenator and illustrating these different perspectives in the new introductory figure 1 which will probably have to be a multipaneled figure to serve eLife's readership). Please also add a short section in the beginning in which the directional terminology used throughout is explained explicitly in the text with clear references to the visual illustration in the new figure 1.

Following the reviewers’ suggestion, we added a paragraph and a figure to introduce the main structures and their definitions in this manuscript and also the anatomical and directional terminology. Particularly, following the reviewers’ suggestion, we used avian terms and added nonavian theropod terms in the parentheses when talking about nonavian theropods. It should be noted that even among birds, the dorsal and ventral surface of coracoid display significantly variable orientation among different avialan groups: in most ornithothoracine birds, the dorsal surface faces dorsocaudally, but in other birds including *Sapeornis*, it faces nearly caudally.

6. To further help eLife's readership and specialist alike, the new figure 1 should also include the following. Comparative images of scapular and coracoid shape in more taxa would, even if they are not based on digital data, which is why we recommend adding these in the new introductory figure 1. E.g. Archaeopteryx, Jeholornis, Confuciusornis are all mentioned but there are no images to help orient the reader and interpret the comparisons that are made due to missing critical visual information that is assumed known – which is not the case for eLife's readership since the journal serves the entire biological (neuroscience, medical as well as several other disciplines including biophysics, biomathematics, bioinspired engineering etc.). This should also be paired with indications of the range of motion that are mentioned in the discussion for greater clarity. The discussion should also mention the limitations in how these ranges of motion have previously been established and how this limits the current analysis in a general biomechanical functional framework. Additionally, visually indicating the attachment of major muscles discussed in the new introductory figure 1, in the context of the body plan, would also be very helpful to orient the reader and follow the discussion. To help the reader further, the introduction, results and Discussion sections should reference to this introductory figure whenever a new concept is discussed in the text (please don't assume the readers find these connections obvious). eLife does not have length restrictions for papers, so the authors can invest words and figures to serve the readers.

Following the reviewers’ suggestion, we added a few more images of several other theropods and illustrated the ranges of motion and the muscles for the readers to have a better understanding of the whole issue, and we also referred to the introductory figure when we talk about the features that are not easily understood by the readers. However, we want to note that we did not illustrate the attachment of all major muscles except the ones we discussed in this manuscript for two reasons: 1. We are afraid to illustrating the muscles that are not discussed in this manuscript might mislead the readers; 2. We have an ongoing project specifically discussing the forelimb motion and the major flight muscles.

7. There seems to be no acknowledged that the morphology of the enantiornithine pectoral girdle in 3D already known from Late Cretaceous specimens, but that the new morphology presentation is clearer for Early Cretaceous specimens, and it can now be recognized that many Late Cretaceous morphologies are also present in Early Cretaceous specimens, which previously could not be recognized without 3D CT data. Please provide this context in the introduction and discussion.

Following the reviewers’ suggestion, we have done appropriate amendments on this.

8. It would be helpful to discuss the homoplasy affecting the evolution of these two bones using non specialist wording, so the general implications are clearer to all readers. Additionally, it would be great to discuss how the absence of an ossified sternum would affect the flight musculature in Sapeornis and what further research could provide more decisive evidence.

Following the reviewers’ suggestion, we added a brief discussion on the major homoplasies present in early-diverging avialans. We add a brief discussion on how the absence of ossified sternum would affect the flight musculature in *Sapeornis*. However, this manuscript focuses on the pectoral girdle morphology in early-diverging avialans and its implication for early flight, and as we noted above, we have a separate project specifically discussing the forelimb motion and the major flight muscles among early-diverging avialans including *Sapeornis*. Hopefully, these two projects together would contribute to a better understanding of early flight.

9. In the discussion, when all the morphological transitions are discussed, citations should be provided for where these have been discussed previously. And where possible, please illustrate them in the new introductory figure 1 so all readers can comprehend the significance of the current research in the literature context.

We have updated all the necessary references.

Reviewer #2 Recommendations for the authors:Line 29 – I am confused with the statement that some taxa have "one area" and others a "double articulation". Could you specify which is "the double articulation" mentioned as widely present among pennaraptorans?

We provided an introductory figure and added a brief description on the general morphology of pectoral girdles among different theropod groups. Simply speaking, the scapula-coracoid articulation has only one connection in all theropods, but this connection is highly localized and display only one surface in enantiornithines and a few crown birds, but it has one main surface and one subsidiary surface in other pennaraptoran theropods including most non-enantiornithine birds (though in these birds, the main surface is more localized, and the subsidiary facet is even smaller compared to non-avialan theropods).

Line 32 – Here produces a sharp cut in the descriptive process, with the treatment of the presence of triosseal canal among birds, without mentioning the (eventual) relation with the "only one vs double articulation" condition analyzed before.

We revised the abstract, and hopefully, the revised version reads better.

Line 33 – Is this "transitional stage" firstly recognized here? Could you precise which are the nodes representing such transition?

Yes, it is. We added a brief statement to improve clarity.

Line 34-37 – I agree with the authors in the conclusions they express in this paragraph. However, I strongly suggest to explain the importance of the "only one vs double scap-cor articulation" and the "transitional stage of triosseal canal" in the anatomical modifications occurred in the line to birds. How their observations on these two aspects modify and expand previous hypotheses on the origin and evolution of bird flight?

Following the reviewer’s suggestion, we revised the abstract.

Line 44 – Let me suggest to cite here Ostrom 1976, which I believe is the founder of this line of studies

Thank you for the references, which are now included in the current draft.

Line 50-51 – Please, check the paper by Imai et al., 2019 on Fukuipteryx. Do these authors already present a 3D reconstruction of the scapular girdle of the kind you present here for other taxa? Anyway, to be the first or not is secondary, and present manuscript represents a formidable progress in our knowledge on the flight apparatus of early birds

This sentence was rephrased

Line 67 – These warnings are necessary and welcome. Many papers on scapulocoracoid in paravian theropods overlook describing the state of preservation of the available materials.

We thank the reviewer for the encouraging comments.

Line 74-75 – I am respectful of the interpretation that current authors are following about the dromaeosaurid affiliation of Rahonavis. However, let me say that Novas et al., (2018. "Postcranial osteology of a new specimen of Buitreraptor". Cretaceous Research) emphasized about the presence of many features that Rahonavis shares with birds more derived than dromaeosaurids and Archaeopteryx (particularly the size and shape of the acromial process!), suggesting that Rahonavis is, in fact, a long-tailed avialan. Interestingly, you are here noting on a quite derived morphology of the acromion, being present in Jeholornis and Rahonavis, thus lending support to the idea that the later taxon is not a dromaeosaurid.

Following the referee’s suggestion, we put a special note on the newly discovered similarity between *Rahonavis* and *Jeholornis.* However, we will not discuss in this manuscript the systematic position of *Rahonavis*.

Line 80 – Is it related with the ventral surface of the acromial process?

No, the main articular surface is not related with the ventral surface of the acromial process.

Line 81 – Let me suggest to indicate also the position with respect to the glenoid. Aside from being "more medially possitioned", is it anteriorly/posteriorly/at level with the glenoid?

Made necessary amendments on this.

Line 111 – Let me suggest to change this part of the phrase, saying something like the following: "…the acrocoracoid process (frequently described as biceps or coracoid tubercle)". The reason is that Archaeopteryx is an avialan sharing a common ancestor with Sapeornis and the rest of the birds, for which the term "acrocoracoid" is used. Afterall is a matter of nomenclature, and all of us agree that such tubercle is homologous.

This sentence was rephrased according to the comment.

Line 113 – Did you check such morphology in other specimens of Sapeornis? Is it natural or is it the result of postmortem compression?

Yes, we did. This feature is also confirmed in 41HIII0405 and IVPP V 19058. We consider that it is a real feature, not the result of compression.

Line 115-118 – Such a fossa on the acrocoracoid process is also seen in Buitreraptor and Rhea, for example (it is something like a volcano with a crater on its top). Novas et al., (2021) interpreted this fossa as the site of attachment of the acrocoraco-acromial ligament. The acrocoracoacromial ligament forms a bridge under which the m. supracoracoideus slides. This ligament is also present in modern flying birds, but it is cranially covered by the omal end of the furcula.Thus, keep in mind this in your considerations that the fossa on the acrocoracoid process served for attachment point for a coracoclavicular ligament connecting the coracoidand furcula (an interpretation that I am not dismissing).

We were aware that both acrocoraco–acromiali and coracoclavicular ligaments are potentially attached to this fossa. However, we did not find any local rugosity, tubercle, or other indicator for acrocoraco–acromiali ligament on the proximal ends of the scapula, which is present in modern birds. Therefore, we reckoned that the acrocoraco–acromiali ligament is absent or weakly developed. On the other hand, as in most flight living birds, the well–developed furcula of *Sapeornis* will maintain the distance between the right and left shoulders of the pectoral girdle. This requires strong ligaments bounded the proximal ends of the furcula and coracoid. Thus, we considered that this fossa is the site of the attachment of the coracoclavicular ligament.

Line 131 – Could you provide an extra figure (a simple line drawing) depicting the correspondence of articular areas? This could be important for other authors in searching for these features in other paravians.

We colored the articular surfaces and the glenoid in Figure 4.

Line 140-141- This is an interesting interpretation. By observing Figure 4 (Sapeornis), the triosseal canal results wide. I wonder if the tendon of the m. supracoracoideus was very thick, or thicker than in other avialans, based on the reconstructed diameter of the canal.

In living birds, the triosseal canal has a large diameter. Some tendons can be surprisingly thick, but some are very thin. Therefore, it is difficult to confirm whether the tendon of *Sapeornis* is thicker than other taxa. If so, it may be related to elevating of the enlarged wing. However, I suspect the M. supracoracoideus tendon is too thin to fill much of the canal. It may slide around within the canal quite extensively.

Line 154-155 – keep in mind that Rahonavis may be a long-tailed avialan, instead of a deinonychosaur.

we choose to not use this reference (*Forster et al., 2020*) and add two new references (Brusatte et al., 2013; Funston et al., 2020) to the statement.

Line 206- Please, insert species name.

We have added “*mongoliensis*” to the text.

Line 237- Let me express again that Rahonavis exhibits many derived features shared with birds more derived than Archaeopteryx, which are absent in deinonychosaurs and unenlagiines. Probably the present manuscript may represents a good place to evaluate the scapular features of Rahonavis, lending support to either deinonychosaur or avialan affinities of this taxon.

Following the referee’s suggestion, we put a special note on the newly discovered similarity between *Rahonavis* and *Jeholornis.* However, we will not discuss in this manuscript the systematic position of *Rahonavis*.

Line 273-275- I agree, of course, with this conclusion. Fossil bones are clear in indicating this condition. I wonder, however, if enantiornithes had (in life) a procoracoidal portion (cartilaginous) for contacting with the well developed acromial region of proximal scapula. A "proto-procoracoid" or more properly a "supracoracoid canal" was present and well developed in Buitreraptor, and presumably served as a "channeling surface" for the m. supracoracoid in its course towards the proximal end of humerus. Crown birds have procoracoid, and a similar channeling surface was present among basal paravians. Enantiornithes may have had such structure, but in an unossified condition.

The coracoid is known as an endochondral bone developing from cartilaginous precursors. As a part of the coracoid, the cartilage procoracoid process will be replaced by bone at the later development stage. So far, no cartilage procoracoid process has been found in adult living birds. Therefore, it is unlikely that Enantiornithes has a cartilage procoracoid process.

Line 290-300- I believe it is unnecessary to repeat. Authors have already clearly explained this in previous lines.

As suggested, we have deleted this paragraph.

Line 313-315 – s it right to say "roofed"? Is the procoracoid forming the caudoMEDIAL wall of the canal, instead of a roof? Apologies if I am wrong.

We changed “roof” to “caudomedial wall”.

Line 322- In Rhea the acrocoraco-acromial ligament bridges the tendon of m. supracoideus, functionally acting as a "triosseal canal". See Novas et al., 2021 for discussions on these asepcts.Line 338-340- I am surprised for this statement: in basal paravians as well as in living birds the m. supracoracoid runs along the LATERAL side of the acromion. The medial surface of the later one is contiguous with the medial surface of scapular blade, thus the medial surface of the acromion contacts with the underlying thorax. The supracoracoid doesn´t run between the scapula and ribs!

Yes, the reviewer is correct. We rejected the assumption of Mayr (2017). The supracoracoideus pulley system in enantiornithine birds was generally the same as that of living birds and other early branches of paravians.

Line 348-349- The presence of a triosseal canal is defined by the contact between the acrocoracoid and epicleidium. If such a contact does not exist, then such a "triosseal" canal is not defined. of course, this is a semantic problem.A completely different matter is the function that this region of the scapular girdle played in paravians: a well defined though is present on the craniomedial surface of coracoid in Buitreraptor (which may have act as a "proto-procoracoid" for chanalizing the m. scapulocoracoid. Thus, the deflection of its tendon was probably operative in basal paravians, and the "inner half" of the triosseal canal was already present even in basal paravians. The shape of the coracoid of Sapeornis (especially the development and position of acrocoracoid) resembles that of Archaeopteryx and Buitreraptor, thus it is expectable that the "pulley system" was postioned and functioned in a similar way in all these forms.In sum, I agree with authors regarding the function of this region of scapular girdle, but the term applied ("triosseal canal") for Sapeornis seems inadequate. Besides, but no less important, a similar pulley-shaped morphology for coracoid was also identified for basal paravians (Buitreraptor, Archaopteryx; Novas et al., 2021).

We disagree that if such a contact does not exist, then such a "triosseal" canal is not defined. As we stated in the previous vision, the triosseal canal is not necessarily formed by all three pectoral elements, and not necessarily a fully enclosed bony passage. The annotation provided for the triosseal canal by Baumel and Witmer (1993) explicitly described variation across taxa in the canal’s architecture.

We believe a partially enclosed structure in non–euornithine birds should be also considered as a triosseal canal as long as it functions as changing the direction of the tendon of the m. supracoracoideus and elevating the wing. In *Sapeornis*, the acrocoracoid process is slightly above the midpoint of the coracoidal glenoid fossa, and consequently above the insertion point during the humeral depression as in the living bird. The vector of the tension is therefore exerted by m. supracoracoideus on the humerus is directed cranially and somewhat dorsomedially. Thus, the term “triosseal canal” is applied for *Sapeornis*.

We agree that acrocoracoid process (coracoid tubercle) are functioned in *Archaeopteryx*, *Buitreraptor*, and other non–avialan pennaraptorans. The term “sulcus m. supracoracoideus” is more appropriate for the groove (or cavity) on the coracoid for the tendon of M. supracoracoideus in these groups.

Line 375-376- Notably, as you shown in Figure 4, Sapeornis does not exhibit a derived condition of acrocoracoid, because it is at level with the glenoid, not in front of it. Thus, I don´t understand why the "extension of acrocoracoid process dorsal to level of coracoidal glenoid fossa" (a condition that seems absent in Sapeornis), is interpreted as characteristic of Step I.

In *Sapeornis* and most other non–ornithothoracine avialans (e.g., *Jeholornis* and *Confuciusornis*), the acrocoracoid process is slightly above the midpoint of the coracoidal glenoid fossa. We have done appropriate amendments in lines 530-571.

Line 403-407- Novas et al., (2020, 2021) said that the pulley was already operative in basal paravians such as Buitreraptor and Archaeopteryx, but the m. supracoracoideus acted as a humeral PROTRACTOR (as in living ratites). This condition is intermediate between the humeral depressor action (in basal theropods) and humeral elevator role (in extanct birds).Novas et al., (2020, 2021) did not say that the "m. supracoracoideus would have acted straightforwardly as a wing depressor".

We are very sorry for the mistake. We have done appropriate amendments in lines 530-571.

Line 419-422- Based on what is beautifully illustrated in present manuscript, the coracoid of Sapeornis does not show a condition of the acrocoracoid sharply different from that in Archaeopteryx. In other terms, the acrocoracoid process is not in front of the glenoid, but almost at the same level. So, I will say the m. supracoracoideus in Sapeornis functioned as a humeral protractor, not as an elevator.

The acrocoracoid process of *Archaeopteryx* is well below the glenoid. We agree that the m. supracoracoideus of *Sapeornis* would have acted primarily as a humeral protractor, but we think it would have pulled the humerus dorsomedially more or less. We have done appropriate amendments on this. please check also throughout the text at lines 530-571.

Line 720 (and labels inserted in Figure 1H)- Is this the border closer to the sternum? Then, it has to be "distal" using the terminology applied on the text (the glenoid represents the "proximal end" of the coracoid).

No, this is the omal end. As we mentioned above, we modified figures to help the readers understand them more easily the morphology.